# Engineering tumor-colonizing *E. coli* Nissle 1917 for detection and treatment of colorectal neoplasia

Bioengineered probiotics enable new opportunities to improve colorectal cancer (CRC) screening, prevention and treatment. Here, first, we demonstrate selective colonization of colorectal adenomas after oral delivery of probiotic *E. coli* Nissle 1917 (EcN) to a genetically-engineered murine model of CRC predisposition and orthotopic models of CRC. We next undertake an interventional, double-blind, dual-centre, prospective clinical trial, in which CRC patients take either placebo or EcN for two weeks prior to resection of neoplastic and adjacent normal colorectal tissue (ACTRN12619000210178). We detect enrichment of EcN in tumor samples over normal tissue from probiotic-treated patients (primary outcome of the trial). Next, we develop early CRC intervention strategies. To detect lesions, we engineer EcN to produce a small molecule, salicylate. Oral delivery of this strain results in increased levels of salicylate in the urine of adenoma-bearing mice, in comparison to healthy controls. To assess therapeutic potential, we engineer EcN to locally release a cytokine, GM-CSF, and blocking nanobodies against PD-L1 and CTLA-4 at the neoplastic site, and demonstrate that oral delivery of this strain reduces adenoma burden by ~50%. Together, these results support the use of EcN as an orally-deliverable platform to detect disease and treat CRC through the production of screening and therapeutic molecules.

Synthetic biology enables the engineering of microbes as living diagnostics and medicines through colonization of niches such as the gastrointestinal tract, skin, lung, and tumors[1–3]. To date, a multitude of studies have shown that bacteria selectively colonize a broad range of tumor types, yet the molecular determinants of this preference for neoplastic tissue are not entirely understood. Existing data suggest tumor-colonizing bacteria leverage specific hallmarks like necrosis, hypoxia and reduced immune surveillance within the tumor microenvironment (TME)[4,5]. Bacteria can further be engineered to produce a range of payloads within the tumor, but studies have thus far focused on intratumoral or systemic administration in subcutaneous tumor models[4,6,7], leading to concerns over toxicity and translational relevance. Moreover, these delivery strategies are limited to palpable lesions, making early intervention challenging. Specific to intestinal disease, oral delivery is a preferred method for probiotic administration, as it enables access to otherwise inaccessible lesions in the gastrointestinal tract. Recent microbial gene circuits have reported the ability of engineered probiotics to sense, record, and respond to signals of inflammation and infection within the gut[8–12]. However, these approaches have thus far relied on known biomarkers and have not yet been used to detect intestinal cancers. As such, synthetic biology tools can be further leveraged to have orally-delivered bacteria encode and locally deliver diagnostic and therapeutic molecules to intestinal lesions.

Colorectal cancer (CRC) is the second leading cause of cancer morbidity and mortality worldwide, with significantly rising incidence rates in younger populations, emphasizing the need for improved and affordable interventions[13]. Colonoscopy is effective at reducing CRC

✉e-mail: susan.woods@adelaide.edu.au; td2506@columbia.edu

incidence and mortality, but is inconvenient, costly, and is associated with rare, but significant, complications[14,15]. Furthermore, genetic conditions that can predispose patients to CRC, such as familial adenomatous polyposis (FAP), result in hundreds of colonic adenomas, the primary precursor lesions of CRC[14,16–18], complicating CRC prevention. While long-term use of non-steroidal anti-inflammatory medications such as aspirin (acetyl salicylic acid) can significantly reduce CRC incidence, protection from conversion to CRC is incomplete[19,20]. Surgical interventions like polypectomy and colectomies are options for early-stage disease and can be used in combination with chemo(radio)therapies[21,22]. Recently, favorable outcomes have been reported in clinical trials with anti-programmed cell death protein-1 (anti-PD1) checkpoint therapy in microsatellite-instability–high (MSI-H) CRC, but with limited success in microsatellite-stable (MSS) CRC disease, which represents ~85% of CRC[23–25]. Thus, there is an unmet need for an approach that can target, detect, and treat adenomas to prevent progression to malignant disease.

Here, we assess whether orally delivered *E. coli* Nissle 1917 (EcN) can selectively colonize adenomas and isolated neoplastic lesions in genetic, orthotopic, and transplant murine models of CRC. EcN is a probiotic strain with demonstrated safety and has been investigated as a chassis for other cancer types[26–28]. We determine the capacity for these bacteria to remain colonized long-term, and the utility of EcN for adenoma detection by recovering EcN from stool or by engineering EcN to produce a small molecule measurable in the urine. Lastly, we investigate whether engineered EcN can deliver immunotherapeutics within adenomas, to manipulate the TME in situ and impact overall disease burden.

## Results

### Adenoma colonization by *E. coli* Nissle 1917 (EcN) in a genetically-engineered mouse model of CRC predisposition

CRC precursor lesions were modeled using *Apc^Min/+* mice[29], a well-established mouse model of FAP, whereby mice spontaneously develop adenomas throughout their intestinal tract. This model is driven by an *APC* gene mutation—an initiating genetic mutation observed in ~80% of human CRC[30] (Fig. 1A). We explored adenoma colonization by orally-delivering EcN encoding a genomically-integrated *luxCDABE* cassette (EcN-lux)[31] to *Apc^Min/+* mice with an intact immune system and microbiome. In vivo imaging of mice dosed with EcN-lux showed elimination of bioluminescent bacteria in healthy wild-type (WT) mouse gut, but retention in the *Apc^Min/+* mouse gut for up to 7 weeks after oral administration (Fig. 1B). Enrichment of EcN-lux in neoplasia was further demonstrated with ex vivo imaging of intestinal tissue, where bioluminescence co-localized with visible macroadenomas and generally, more bioluminescence was observed in the distal small intestine where adenoma burden was the greatest (Fig. 1C). Subsequent plating of homogenized intestinal tissue on antibiotic-selective Luria broth (LB) plates specific for EcN-lux, indicated that no detectable EcN-lux could be recovered from wild-type mouse tissue, suggesting that a long-term niche is not formed in the gut unless neoplastic tissue is present (Fig. 1D). To further investigate bacteria localization, EcN-lux was engineered to release a human influenza hemagglutinin (HA)-tagged protein under control of a lysis circuit[32,33] and orally delivered to *Apc^Min/+* mice. Four weeks after oral delivery, mice were sacrificed and their intestinal tissue was probed for positive HA signal. Dark positive stains of EcN-tagged payloads were identified in adenomas of varying sizes, suggesting functional delivery

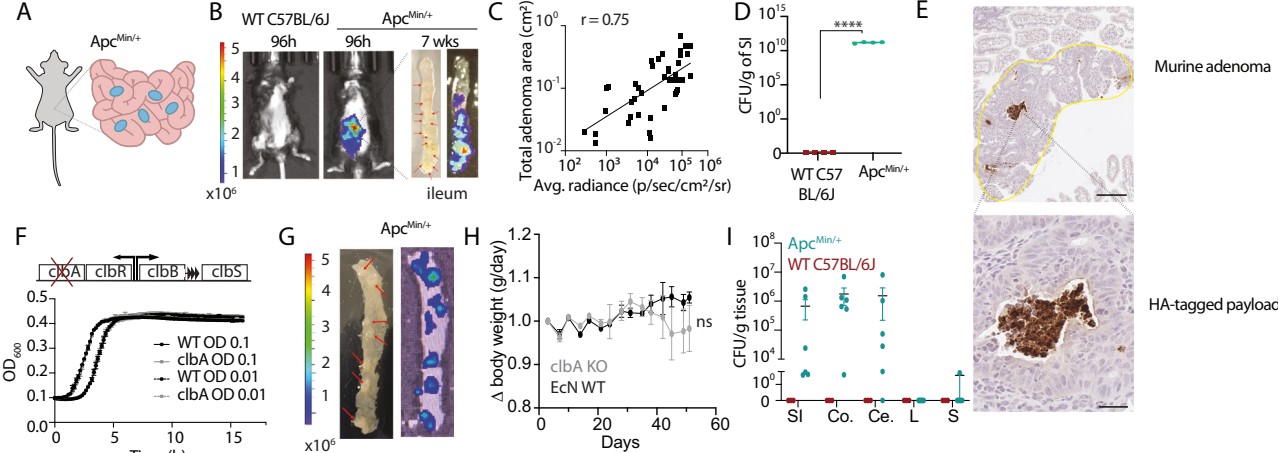

**Fig. 1 | Adenoma colonization of *E. coli* Nissle 1917 (EcN) in a genetically-engineered mouse model of CRC. A** Schematic of spontaneous intestinal adenomas in *Apc^Min/+* model. 12-week-old *Apc^Min/+* mice were gavaged twice, 3–4 days apart with EcN expressing luxCDABE luciferase cassette (EcN-lux). **B** EcN-lux was visualized using an IVIS for bioluminescence in vivo 96 h post dosing. After 7 weeks, mice were sacrificed, intestinal tissue was excised and imaged ex vivo for bioluminescence. Red arrows point to macroadenomas on distal intestinal tissue (*n* = 20 WT, *n* = 25 *Apc^Min/+*). **C** Plot where x-axis is the bioluminescence signal measured from ex vivo images of dissected intestinal tissue and y-axis is the total adenoma area in matched tissue sections as measured in H&E-stained images (*n* = 35 intestinal sections, r = 0.75, Spearman correlation coefficient). **D, E** In separate cohorts, mice were dosed with EcN-lux or EcN producing an HA-tagged protein and **D** sacrificed at 1 week post dosing. Intestinal tissue was homogenized and plated on antibiotic-selective plates for EcN-lux to quantify colony-forming-units (CFU) per gram of tissue (*n* = 4 WT, *n* = 4 *Apc^Min/+*, ****p < 0.0001) or **E** sacrificed at 4 weeks post dosing and intestinal tissue was paraffin-embedded, sectioned and stained by anti-HA

immunohistochemistry. Dark brown stain depicts HA-tagged protein produced by EcN in adenomas. Scale bars represent 200 μm (top) and 50 μm (bottom). **F** Schematic (Top) of colibactin-encoding operon in EcN whereby *clbA* is knocked out and colibactin production is disrupted. Plate reader experiment of strain variant growth kinetics at 0.1 and 0.01 seeding OD. **G–I** 12-week-old *Apc^Min/+* mice and WT littermates were gavaged twice, 3–4 days apart with 10^9 CFU bioluminescent EcN-lux or EcNΔ*clbA*-lux. **G** After 1 week, mice were sacrificed, intestinal tissue was excised and ex vivo imaged for bioluminescence. Red arrows point to macro-adenomas on distal small intestinal tissue (representative image from *n* = 5 mice). **H** Body weight of both EcN-lux and EcNΔ*clbA*-lux-treated mice were tracked over the course of the experiment (*n* = 3 EcN-lux, *n* = 4 EcNΔ*clbA*-lux, two-way ANOVA, ns, not significant). **I** In a separate cohort one week post dosing, mice were sacrificed, small intestine (SI), colon (Co), cecum (Ce), liver (L), and spleen (S) were harvested, homogenized, and plated on antibiotic selective plates to recover EcNΔ*clbA*-lux per gram of tissue (*n* = 3 WT, *n* = 6 *Apc^Min/+* mice per group). All error bars represent S.E.M. Source data are provided as a Source data file.

of payloads released by the lysis circuit following oral delivery of the engineered strain (Fig. 1E, Supplementary Fig. 1).

Due to increased concerns of colibactin-producing bacteria like EcN being pro-carcinogenic[34], we disrupted colibactin production by deleting the *clbA* gene (EcN*ΔclbA*)[35,36] and observed no changes in bacterial growth kinetics at multiple seeding densities compared to the parent EcN strain (Fig. 1F), aligning with previous studies[37]. Similar to above, we explored neoplasia colonization by orally-delivering EcN*ΔclbA*-lux to *Apc^{Min/+}* mice and observed co-localization of bioluminescent bacteria with macroadenomas (Fig. 1G), suggesting that colonization does not rely on the presence of the *clbA gene* or an intact colibactin-encoding operon. In *Apc^{Min/+}* mice treated with either the EcN-lux or EcN*ΔclbA*-lux strain, we did not observe any notable differences in mouse body condition and weight, as others have demonstrated previously with EcN strains in other mouse types[38–40] (Fig. 1H). Forty-eight hours post dosing, the liver, spleen, cecum, colon, and the small intestine from each mouse were harvested and plated on antibiotic-selective LB plates to recover EcN*ΔclbA*-lux. While EcN*ΔclbA*-lux was found abundantly in the intestines of polyp bearing animals compared to healthy controls, fewer to no bacteria were recovered from the liver and spleen of *Apc^{Min/+}* mice or from wild-type control mice, demonstrating minimal off-target localization (Fig. 1I).

## Tumor colonization by EcN in orthotopic mouse models of CRC

We next tested selective colonization of isolated lesions by evaluating EcN-lux in two orthotopic models of CRC, representative of MSS and MSI subtypes of disease, whereby murine CRC organoids were injected into the distal colon of recipient mice and tumor grade tracked via weekly colonoscopy[41] (Fig. 2A, Supplementary Fig. 2). To begin, MSS CRC tumor-bearing mice were pre-treated with broad-spectrum antibiotics to disrupt the normal microbiota composition, a common phenomenon in gastrointestinal diseases including CRC[42]. EcN-lux was orally delivered and in vivo imaging five days post dosing revealed co-localization of bioluminescent EcN-lux with colon tumors (Fig. 2B, C). Subsequent homogenization and plating of excised organs on antibiotic-selective LB plates confirmed EcN-lux was significantly enriched in tumors compared to adjacent healthy tissue and peripheral organs (Fig. 2D). The median diameter of EcN-lux colonized tumors was 2 mm (+/−1.2 mm), suggesting the size of neoplastic lesions detected using this EcN-lux platform was similar to the size of diminutive (0–5 mm) polyps in humans[43]. Specific localization of EcN-Lux in these tumors by RNA in situ hybridization, suggested that the bacteria can predominantly be found in nests on the luminal tumor surface and can co-locate with hypoxic tumor regions that may further facilitate bacterial growth within the tumor space (Fig. 2E, F, Supplementary Fig. 3). Similarly, oral EcN-lux dosing of a MSI CRC model (Fig. 2G), without antibiotic pretreatment, resulted in selective, significantly enriched colonization of tumors in comparison to adjacent tissue and organs as measured by ex vivo luminescence imaging and CFU plating (Fig. 2H, I).

## Tumor colonization of EcN in human CRC

To explore the translational potential of our pre-clinical findings for humans, we first determined whether gram-negative bacteria, such as *E. coli*, could be visualized in tumor samples from CRC patients using a pan-gram negative lipopolysaccharide (LPS) bacterial stain. Consistent with previous reports of the tumor-associated CRC microbiome, we observed LPS+ bacteria associated with human CRC (Supplementary Fig. 4)[44–46]. We subsequently undertook a clinical trial to specifically examine EcN colonization in CRC patients (Supplementary Table 1). A commercially available, non-genetically modified form of EcN, Mutaflor, or placebo, was orally administered to CRC patients for two weeks, prior to tissue resection. A small number of participants were recruited to this study after early closure due to COVID-19-related trial restrictions and concerns over colibactin-producing *E. coli*, such as

EcN[34]. Homogenates from matched normal and tumor tissue (*n* = 8 patients) were cultured to enrich for microbial content, DNA was then isolated and subjected to qPCR assays (Supplementary Fig. 5). Despite the smaller than intended number of samples, EcN-specific PCR amplicons[27,47] indicated significant enrichment of this bacteria in cultures from tumor tissue in patients administered Mutaflor, but not placebo controls (Fig. 2J, K, Supplementary Fig. 5C).

## Engineering a stool and urine-based EcN platform for non-invasive adenoma tracking

The phenomenon of EcN neoplasia-selective colonization suggested its utility as a platform to monitor adenoma presence. Since stool-based tests are a common non-invasive screening tool available for CRC, we first determined if EcN-lux recoverable in stool could be used to non-invasively monitor the presence of adenomas over time. To do this, we orally dosed both healthy wild-type (WT) and *Apc^{Min/+}* mice with EcN-lux, collected stool pellets at predetermined timepoints, homogenized and lastly plated fecal matter on antibiotic-selective LB agar plates (Fig. 3A). During the first 7 h post dosing, both WT and *Apc^{Min/+}* mice had comparable shedding of EcN-lux CFU in their stool, corresponding to material transit time through the gut[48]. However, by 24 h, levels of EcN-lux were undetectable in some healthy mice and by 48 h we were unable to recover EcN-lux from healthy mouse stool, but EcN-lux was still recoverable from *Apc^{Min/+}* mouse stool (Fig. 3B). We observed a similar significant retention of EcN-lux in fecal samples in neoplasia bearing, compared to normal control mice, using the MSI CRC model or the *Apc^{Min/+}* model dosed with the EcN*ΔclbA*-lux mutant strain (Fig S6).

While this stool test could be a useful method for adenoma tracking, we aimed to investigate a more accessible diagnostic readout. To do this, we engineered EcN to produce a molecule that could be conveniently recovered from bodily fluids. We chose to encode the production of salicylate due to its role in CRC chemoprevention[19,20] and because it can be feasibly detected in urine[49]. To optimize salicylate production we engineered a library of strain variants to express genes critical to the shikimate pathway, which is responsible for converting endogenous bacterial chorismite to salicylate[50,51]. Specifically, genes *mbti, irp9, menF, entC*, or *pchA* were cloned onto plasmids also encoding *pchB* with either low (sc101*) or high (ColE1) copy number origins. These plasmids were then cloned into two distinct EcN strains −the EcN^{ATT}*ΔclbA*-lux (EcN) mutant or the EcN^{ATT}*ΔclbA*-lux (EcN^{ATT}), where the latter included genomically integrated *aroG*, *tktA*, and *talB* genes. As these genes are involved in the shikimate and pentose phosphate pathways, we hypothesized that their integration would redirect metabolic flux towards increased salicylate production (Fig. 3C). We observed that some variants containing the higher copy plasmid were unable to grow or contained mutations in the gene of interest, which may be due to the toxic effect of higher levels of salicylate on bacterial cell viability (Supplementary Fig. 7). Proceeding with the viable EcN^{ATT} strains, we used liquid-chromatography mass spectrometry (LC-MS) to probe for salicylate in supernatant collected from overnight cultures. Generally, higher copy variants produced more salicylate with EcN^{ATT}-EntC-ColE1 producing ~15 μM per 10^9 bacteria (Fig. 3D). Furthermore, comparison of the highest producing strains from both copy number variants, sc101*-Irp9 and EntC-ColE1, demonstrated higher salicylate production when encoded by the EcN^{ATT} mutant compared to the EcN strain (Fig. 3E). Taken together, we determined that the highest producing salicylate variant was the EcN^{ATT}-EntC-ColE1 strain.

To evaluate this optimized strain's diagnostic potential, we dosed *Apc^{Min/+}* and WT mice with EcN^{ATT}-EntC-ColE1 strain and collected both stool and urine at predetermine timepoints. EcN recovered from stool collected at 48 h post gavage was used to confirm colonization or lack thereof of the strain in *Apc^{Min/+}* and WT mice. Furthermore, stool was also plated on kanamycin plates selective for the salicylate-encoding

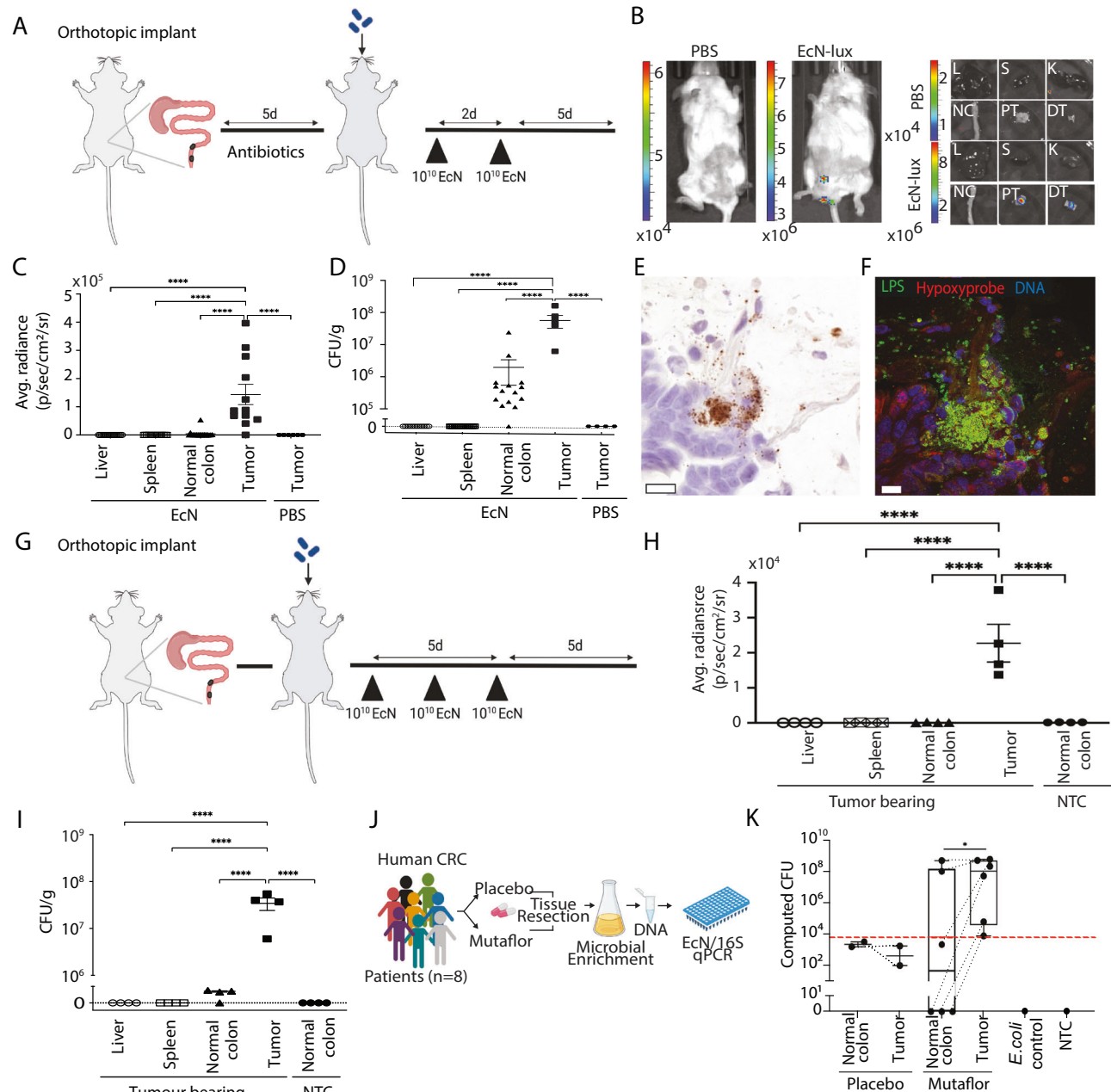

**Fig. 2 | Tumor colonization of *E. coli* Nissle 1917 (EcN) in orthotopic mouse models and human CRC patients. A** Schematic of experimental timeline for MSS CRC model. Tumor growth was monitored by colonoscopy, with animals orally dosed twice with EcN-lux or PBS. **B** Imaging 5 days after last dose for bioluminescence, with **C** luminescence quantified in organs ex vivo (L, liver, S, spleen, and NC, normal colon *n* = 17, PT, proximal tumor, DT distal tumor, *n* = 12, PBS tumors *n* = 6). **D** Tissues were homogenized, plated on antibiotic-selective plates, and quantified for CFU per gram (*n* = 17 from EcN-dosed non-tumor mice and EcN-dosed tumor mice *n* = 6, PBS-dosed tumor mice *n* = 4). **E, F** Representative images of orthotopic CRC from mice dosed as (**A**), with serial sections showing (**E**) EcN-lux specific location by RNAscope in situ hybridization for *lux* (brown, scale bar 20 µm) and (**F**) immunofluorescent staining for Hypoxyprobe (red) and lipopolysaccharide (LPS, green, scale bar 10 µm, *n* = 5 mice). **G** Schematic of experimental timeline for MSI CRC model. Tumor and non-tumor bearing control animals orally dosed thrice with EcN-lux. **H** Imaging 5 days after last dose with luminescence quantified in organs ex vivo and **I** tissues homogenized, plated on antibiotic-selective plates, and

quantified for CFU per gram (*n* = 4 for both tumor bearing and no-tumor control groups). **J** Schematic of human clinical trial. **K** Matched normal and tumor tissue homogenates from CRC patients administered placebo (*n* = 2) or Mutaflor (trade name for EcN) (*n* = 6) for 2 weeks. Tissue samples were used to inoculate liquid culture for microbial enrichment. DNA isolated from culture was subjected to qPCR with an EcN specific assay. Mean value of 4 technical replicates shown per sample, box in the plot minima–maxima is 25th–75th percentile, whiskers at lowest and highest values, bar at median per group. Red dashed line depicts the lower limit of detection of each assay based on standard curve dilution series, dot points above the line have detectable PCR amplicon signal. No EcN signal was detected in E. coli control (ATCC 9522), no template (NTC), or buffer only DNA prep controls. **C, D, I** ****p < 0.0001, one-way ANOVA with Tukey's multiple comparisons. **H** ****p < 0.0001, two-way ANOVA, Fischer's LSD test). **K** *p = 0.03, two-way paired t-test. Figure 2J was created with BioRender.com. Source data are provided as a Source data file.

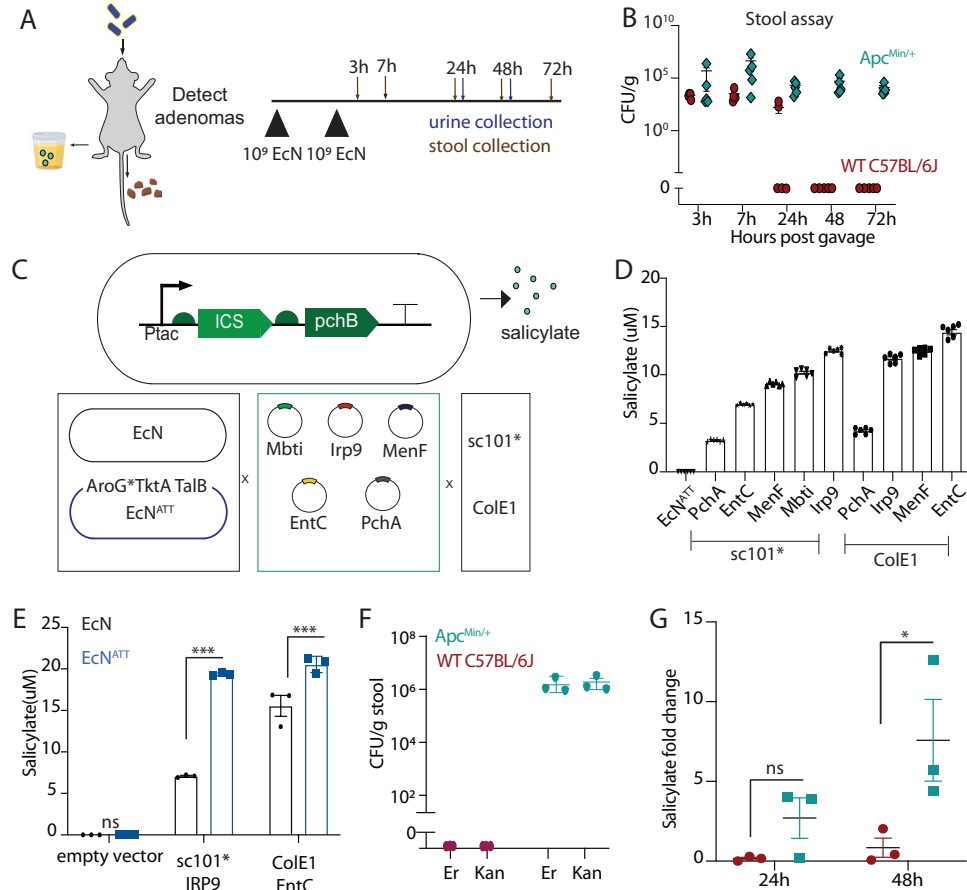

**Fig. 3 | Engineering of stool and urine-based EcN platform for non-invasive adenoma tracking. A** Schematic of orally-delivered EcN probiotic to be detected in fecal matter by quantifying colony-forming units or urine by metabolite quantification. **B** 15-week-old $Apc^{Min/+}$ mice were dosed orally 3–4 days apart with EcN and one stool pellet was collected at 3, 7, 24, 48, and 72 h after last dose, homogenized, plated on antibiotic-selective plates and quantified for CFU ($n = 5$ mice per group). **C** Schematic of the genetic construct where the isochoristmate synthase (ICS) and *pchB* genes result in salicylate production (Top). These enzymes were cloned onto plasmids and transformed into EcN or a metabolically engineered EcN strain (EcN$^{ATT}$). **D, E** Overnight cultures of these strain variants were optical density-matched and LC-MS was used to probe for salicylate presence between **D** plasmid-

encoded EcN$^{ATT}$ variants ($n = 6$ biological replicates per group) and **E** selected variants between the EcN and EcN$^{ATT}$ strains ($n = 3$ biological replicates per group, two-way ANOVA, Holm-Sidak post test ***$p = 0.0001$, ****$p < 0.0001$). All samples were normalized to an internal isotope-labeled D4-salicylate standard. **F, G** 15-week-old $Apc^{Min/+}$ mice were dosed with $10^9$ EcN$^{ATT}$-EntC-ColE1, **F** stool was collected at 48 h, homogenized, and plated on selective plates to quantify recoverable bacteria ($n = 3$ mice per group, Er) and recoverable bacteria retaining the salicylate-encoded plasmid (Kan) and **G** urine was collected 24 h and 48 h after dosing and salicylate quantified using LC-MS in WT and $Apc^{Min/+}$ mice ($n = 3$ mice per group, ns = not significant, *$p = 0.0228$ two-way ANOVA with Holm-Sidak post test). All error bars represent S.E.M. Source data are provided as a Source data file.

plasmid to confirm plasmid retention (Fig. 3F). Urine was collected before dosing to establish a baseline and then again at 24 and 48 h after dosing. LC-MS was used to probe for salicylate presence and we observed that $Apc^{Min/+}$ mice had up to five times more salicylate 48 h after dosing compared to baseline levels, whereas salicylate levels in WT mice did not change over time (Fig. 3G). In separate studies, increased salicyluric acid—the primary metabolite of salicylate—was also detected at higher levels in urine of $Apc^{Min/+}$ mice (Supplementary Fig. 8). Taken together, these data suggest that engineered bacteria can be delivered to neoplasia-bearing mice, maintain their genetic circuitry, and be used as a proxy for non-invasive adenoma tracking and possible early detection in both fecal matter and urine-based assays.

### Treatment with EcN engineered to produce immunotherapeutics reduces adenoma burden and modifies the tumor-immune microenvironment

With the ability to non-invasively determine adenoma presence, we next sought to address whether our screening system could be adapted for therapeutic purposes and reduce polyp burden. As the $Apc^{Min/+}$ model is considered to be microsatellite stable (MSS), which

traditionally responds poorly to immunotherapeutic approaches, we hypothesized that the bacteria in our probiotic platform would serve as an immune adjuvant and also a chassis to deliver multiple immunotherapeutic payloads. We combined therapeutic delivery with an EcN-lux strain genomically-encoded with a lysis circuit optimized (SLIC) to maximize immunotherapeutic release and also aid in biocontainment by controlling EcN populations[32,33]. Drawing upon previous work[32,33], use of this lysis-based release mechanism is needed for effective release of therapies and is critical for therapeutic efficacy. Furthermore, bacterial lysis results in immune adjuvants that further enhance therapeutic effects of immunotherapy. SLIC was used to deliver nanobodies blocking PD-L1 and CTLA-4 targets and cytokine GM-CSF (SLIC-3), which we have previously demonstrated enhance efficacy of checkpoint blockade therapy in a subcutaneous mouse colorectal model when delivered intratumorally[33] (Fig. 4A). Here, the $Apc^{Min/+}$ mice were either orally dosed twice within 3–4 days with PBS or SLIC-3 and then sacrificed ~1 month later. Histological analysis of hematoxylin and eosin-stained tumors demonstrated an overall reduction of adenoma area (Fig. 4B) and number (Supplementary Fig. 9A, B) by ~47% with SLIC-3 treatment. Notably, an increased percentage of smaller adenomas was observed in SLIC-3-treated mice,

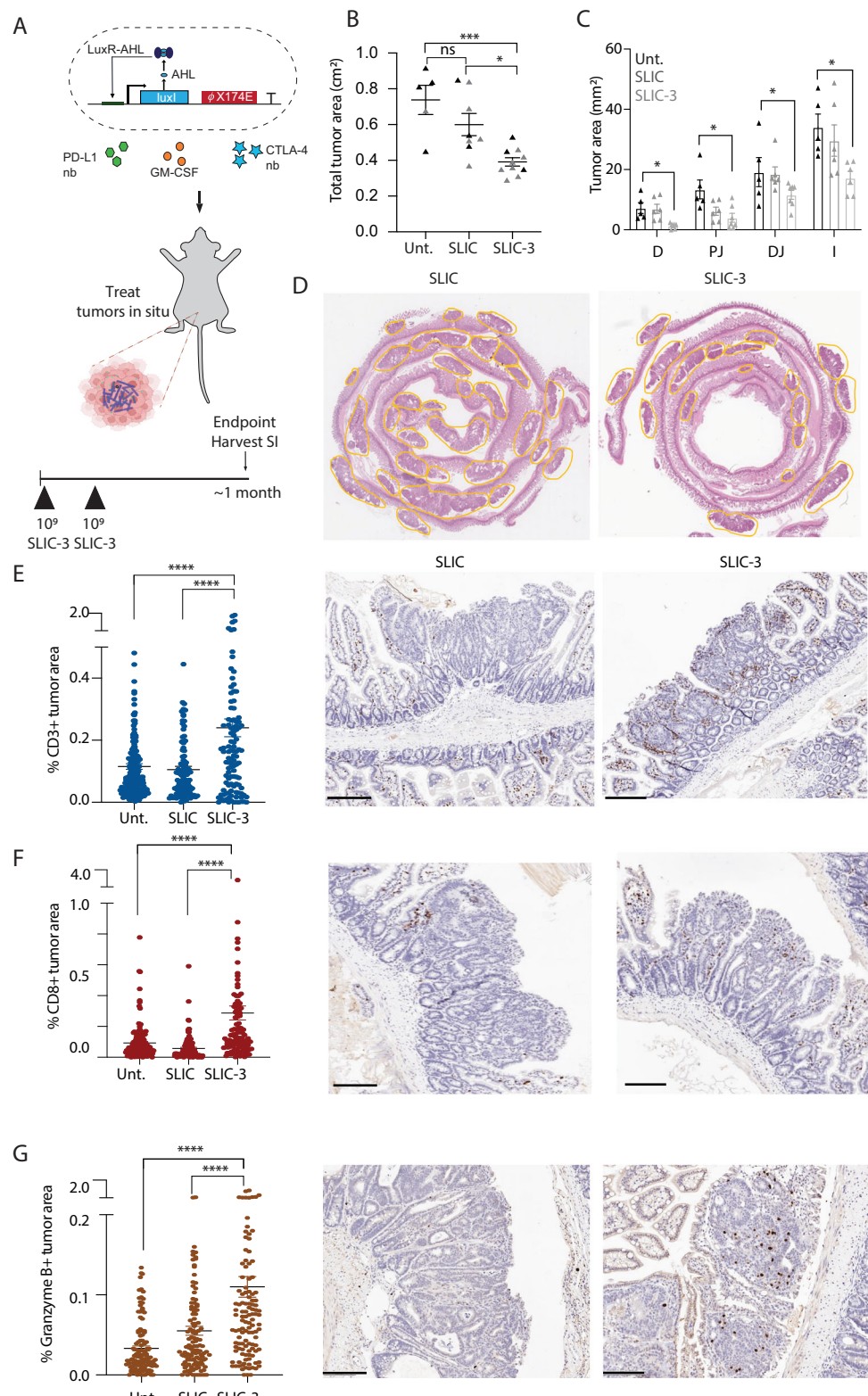

whereas PBS-treated mice tended to have larger adenomas (Supplementary Fig. 9C). Moreover, this reduction was not specific to a location and was observed throughout the small intestine (Fig. 4C, D). Interrogation of immunophenotype on tissue sections suggests that reduction in adenoma burden is associated with increased infiltration of CD3+, CD8+ cells and production of granzymeB within adenomas, suggesting immune-mediated tumor cell killing in SLIC-3 treated mice (Fig. 4E–G, Fig. Supplementary Fig. 10). Moreover, there is a trend

whereby SLIC alone increased granzymeB staining when compared to untreated mice, likely due to the inherent immunogenicity of lysed bacteria, underlying the added benefit of leveraging a bacteria-based platform[33,52].

## Discussion

Altogether, we have demonstrated selective and robust colonization of adenomas and tumors in two distinct orthotopic murine models and

**Fig. 4 | Treatment with EcN engineered to produce immunotherapeutics reduces adenoma burden and modifies the tumor-immune microenvironment. A** Schematic of orally-delivered EcN probiotic engineered to lyse and produce immunotherapeutic proteins in situ (top) and schematic of dosing regimen (bottom). **B, C** 15-week-old *Apc^Min/+* mice were dosed twice within 3–4 days and then with PBS (Unt), EcN genomically encoding a lysis circuit (SLIC) or SLIC producing granulocyte-macrophage colony-stimulating factor (GM-CSF), and blocking nano-bodies against PD-L1 and CTLA-4 targets (SLIC-3). One month after dosing, mice were sacrificed, intestines were bisected, swiss-rolled, paraffin-embedded, sectioned, stained with hemotoxin and eosin and quantified for **B** overall tumor area (black: female mice; gray: male mice), *n* = 5 mice (Untreated), *n* = 8 (SLIC), *n* = 10 (SLIC-3) and **C** tumor area along the intestine, D, duodenum, PJ, proximal jejunum, DJ, distal jejunum, I, ileum (**p* = 0.0215, ***p* = 0.0008, ****p* = 0.001, ns, not significant, ordinary one-way ANOVA test with Holm-sidak multiple comparisons test,

*n* = 5 mice (Untreated), *n* = 7 (SLIC), *n* = 7 (SLIC-3). **D** Representative H&E-stained histology images of SLIC and SLIC-3 treated mice. **E–G** Using immunohistochemical (IHC) techniques, intestinal tissue sections from each mouse groups were stained and quantified for **E** CD3+, *n* = 3 mice, 113 polyps (SLIC-3), *n* = 4 mice, 107 polyps (SLIC), *n* = 3 mice, 131 polyps (Untreated), *****p* < 0.0001 ordinary one-way ANOVA with Tukey post-test, **F** CD8+, *n* = 3 mice, 111 polyps (SLIC-3), *n* = 4 mice, 141 polyps (SLIC), *n* = 3 mice, 112 polyps (Untreated), *****p* < 0.0001 ordinary one-way ANOVA with Tukey post-test, and **G** granzymeB+ cells, *n* = 3 mice, 124 polyps (SLIC-3), *n* = 3 mice, 120 polyps (SLIC), *n* = 3 mice, 112 polyps (Untreated), *****p* < 0.0001 ordinary one-way ANOVA with Tukey post-test). Representative IHC images of all three markers are shown beside their respective plots with positive staining depicted as brown puncta in SLIC and SLIC-3-treated mice. Scale bars represent 200 μm. All error bars represent S.E.M. Source data are provided as a Source data file.

human CRC patients with orally delivered probiotic EcN. Leveraging this colonization ability, we demonstrated the possibility for engineered EcN in adenoma diagnosis through non-invasive stool and urine assays. Furthermore, we demonstrated therapeutic potential by encoding EcN to produce checkpoint inhibitor therapies and cytokine GM-CSF, thereby significantly reducing adenoma burden in a model of MSS CRC through oral delivery, a disease subtype that in humans is normally unresponsive to systemically-administered checkpoint inhibitors[53,54]. Moreover, additional benefits were observed in using a probiotic platform compared to conventional checkpoint therapies, including possible remodeling of the TME. Interestingly, T cells were detected at the periphery of polyps in the untreated group, consistent with multiple studies demonstrating a correlation between WNT/beta-catenin activation characteristic of lesions in *Apc^Min/+* mice and T cell exclusion[55,56]. These data also demonstrate enhanced T cell infiltration into the adenoma core upon bacterial-production of GM-CSF and checkpoint inhibitor nanobodies, potentially overcoming WNT/beta catenin-mediated T cell exclusion; however, more experimentation is necessary to understand the underlying mechanisms involved. In addition, therapeutic efficacy of SLIC-3 is limited, but as our probiotic platform is modular, there is the possibility to expand both screening and therapeutic cargo to explore other therapeutic combinations and achieve enhanced adenoma reduction. Finally, evaluation in expanded cohorts for both screening and diagnostic applications is needed prior to translation of these technologies.

In the absence of neoplasia and consistent with our results, orally delivered EcN is not a persistent colonizer of the healthy mouse gut[2,57], nor inflammatory lesions present in a mouse model of inflammatory bowel disease[58]. Early biodistribution studies with intravenously administered EcN reported that while EcN transiently localized to skin wounds in mice, colonization was not observed at inflamed cutaneous sites or once wounds had healed[59] suggesting that persistent colonization is a specific feature of neoplasia. While we explored colonization of EcN across multiple murine models both with intact and disrupted microbiomes, more investigation into the effects of the native tumor microbiome and gut disorders on EcN colonization could provide insight into generalizability of this approach across patients with varying symptoms, diets, and microbiota, including those who might already have baseline levels of EcN present[60]. Equally, in our trial two of the six participants in the Mutaflor arm of the trial had comparatively high levels of EcN in the normal, adjacent colon samples and did not show EcN enrichment in tumor samples over normal. As the stool CFU data from our mouse models (Figs. 3B, S6) indicate that individual mice show variability in the time required to clear EcN from their gastrointestinal tract in the absence of neoplasia after EcN dosing, this result in our human cohort may be due to the absence of a multi-day wash-out period in our trial protocol and/or inter-patient variability in colonization by probiotic bacteria[60]. As such, selective colonization of neoplastic tissue in humans after oral administration of EcN warrants further investigation in expanded cohorts. Of note, safety concerns

regarding EcN, as a possible colibactin-producing *E. coli* strain, remain. Future trial design should incorporate colibactin-mutant strains that retain neoplastic colonization properties (Fig. 1). The limitations of our current trial include: small participant numbers; incomplete previous antibiotic usage data; a reliance on self-reporting of probiotic administration by trial participants; in the absence of other highly sensitive and EcN-specific markers to enable tissue localization, the detection of non-genetically modified EcN was only able to be assessed using PCR; lastly, a wash-out period post-probiotic treatment, but prior to tissue resection, was not included but will be key to understanding the longevity of EcN neoplastic colonization. Nevertheless, this data demonstrates EcN can colonize human colorectal neoplasia and thus supports the findings from our mouse studies.

In addition, while mechanisms such as selective adhesion to tumor antigens and attraction to small molecules produced in tumors that drive translocation of bacterial pathogens across the gastrointestinal barrier, are known for other *E. coli* strains, *Salmonella typhimurium*, and *Fusobacterium nucleatum*[61–63], a deeper understanding of characteristics governing EcN establishment within tumors is needed. The presented platform demonstrates that engineered probiotic strains maintain their programmed behavior within the complex gut environment and highlights their potential for a range of diagnostic and therapeutic applications. Taken together, these data lay the groundwork for future pre-clinical and clinical testing of engineered EcN for early CRC detection and treatment.

## Methods

### Strains and plasmids
All bacterial strains used were luminescent (integrated *luxCDABE* cassette) so they could be visualized with the In Vivo Imaging System (IVIS). The EcNΔ*clbA* strain was engineered using the lambda-red recombineering method[64]. The salicylate-encoding plasmid was constructed using Gibson assembly methods or restriction enzyme-mediated cloning methods whereby isochorismate synthase genes (*irp9, mbtI, menF, entC,* and *pchA*) and *pchB* genes were cloned onto medium or high-copy origin plasmids and driven by the *tac* promoter. Pathway-engineered EcN was constructed by integrating *aroG^fbr*, *tktA*, and *talB* genes using pSPIN plasmid. The SLIC and SLIC-3 strains were constructed as previously described[33].

### Bacterial preparation for oral administration
Overnight cultures of EcN-lux were diluted 1:100 into LB with 50 ng/ml erythromycin and cultured to an OD600 of 0.1–0.5 on a shaker at 37 °C. Bacteria were collected by centrifugation at 3000–5000 × *g*, washed three times with sterile PBS, resuspended in sterile ice-cold PBS with a total of 100–200 μL dosed orally at a concentration of ~10^10–10^11 CFU/ml. Salicylate-producing strains were cultured similarly with an additional 50 μg/ml kanamycin added to retain the salicylate-encoding plasmid. SLIC strains were prepared as previously described[33]. Briefly, growth media for SLIC and SLIC-3 strains also

contained 0.2% glucose to suppress premature lysis in culture. In addition, SLIC-3 strains were grown with 50 μg/ml kanamycin.

## Organoid culture

Mouse CRC $Braf^{V600E};Tgfbr2^{\Delta/\Delta};Rnf43^{\Delta/\Delta}/Znrf3^{\Delta/\Delta};p16$ $Ink4a^{\Delta/\Delta}$ ($Braf^{V600E}\Delta TRZI$, MSS CRC) and $Apc^{\Delta/\Delta}$, $Kras^{G12D/\Delta}$, $Trp53^{\Delta/\Delta}$, $Mlh1^{\Delta/\Delta}$ (AKPM, MSI CRC) organoids were generated using CRISPR/Cas9 genome engineering and expanded for injection into mice in matrigel culture as described[41]. Culture medium was Advanced Dulbecco's modified Eagle medium/F12 (Life Technologies) supplemented with 1x gentamicin/antimycotic/antibiotic (Life Technologies), 10 mM HEPES, 2 mM GlutaMAX, 1xB27 (Life Technologies), 1xN2 (Life Technologies), 10 ng/ml human recombinant TGF-β1 (Peprotech). Further media supplementation with 50 ng/ml mouse recombinant EGF (Peprotech) for MSS organoids, or 100 ng/ml mouse recombinant noggin (Peprotech), 50 μM Nutlin (Sigma) and 1 mM EGFR inhibitor (Sigma-Aldrich) for MSI organoids. Immediately after each split, organoids were cultured in 10 μM Y-27632 (In Vitro Technologies), 3 μM iPSC (Calbiochem Cat #420220), 3 μM GSK-3 inhibitor (XVI, Calbiochem, # 361559) for the first 3 days.

## Orthotopic mouse models of CRC

All animal experimentation involving the orthotopic CRC implant models was approved by the institutional animal ethics committee of the South Australian Health and Medical Research Institute (SAHMRI) (SAM-319, SAM-20-031, SAM-21-041). Orthotopic injections to generate distal colon tumors were undertaken as previously described[41]. In brief, $NOD.Cg$-$Prkdc^{scid}Il2rg^{tm1Wjl}/SzJ$ (NSG) mice (male and female, 6–12 weeks old, recipients for MSS organoids) or C57BL/6 mice (male, 6–12 weeks old, recipients for MSI organoids) were obtained from the SAHMRI Bioresources facility and housed under SPF conditions. Digested MSS or MSI organoid clusters (equivalent to ~150 organoids) were resuspended in 20 μL 10% GFR matrigel 1:1000 India ink, 10 μM Y-27632 in PBS and injected into the mucosa of the distal colon of anaesthetized mice using colonoscopy-guided orthotopic injection (2 injection sites/mouse). Injection sites were monitored by weekly colonoscopy. EcN administration began once the tumors were clearly established at grade 3 to 4 on the Becker scale[65], 4–6 weeks post organoid injection (Fig S4). Mice bearing MSS tumors were treated with broad-spectrum antibiotics to generate gut dysbiosis through administration of 0.5 g/L neomycin and 1 g/L ampicillin ad libitum in drinking water for 5 days. This was halted 6 h prior to EcN administration. All mice were regularly monitored for signs of clinical deterioration (such as body condition, absence of stool production, weight loss) and euthanized if clinical score reached 3, or timed endpoint 5 days after last EcN administration.

## $Apc^{Min/+}$ mouse model of CRC

All animal experimentation related to the $Apc^{Min/+}$ mouse model of CRC was approved by the Institutional Animal Care and Use Committee (Columbia University, protocols AC-AAAN8002 and AC-AAAZ4470). All mice were regularly monitored and euthanized based on veterinarian recommendation or when they reached ~20 weeks of age. In all therapeutic studies wild-type littermates of $Apc^{Min/+}$ on the C57BL/6 background were used and both males and females were treated and evenly distributed among groups. For diagnostic studies, $Apc^{Min/+}$ mice were purchased from Jackson Laboratories and purchased C57BL/6 mice were used as wild-type controls.

## Bioluminescence imaging

To quantify the EcN-lux derived luciferase signal in our mouse models of CRC we used a Xenogen in vivo imaging system (IVIS) Spectrum Imager (Perkin Elmer Inc). Following necropsy, individual mouse tissues were collected into individual wells of a 6-well plate, weighed and background (stage alone) subtracted average radiance (photons/s/

$cm^2$/sr) measurements were used to correct for the area being measured which differed for each tissue analyzed.

## Colony-forming unit assays

Excised mouse tissues were placed aseptically into 5 ml 20% glycerol in PBS and homogenized in MACS Gentle cell dissociator C tubes, one tissue per tube using program C. 100 μl of each tissue homogenate glycerol stock was serially diluted 1:100 six times. 10 μl of each dilution was spotted onto an LB agar plate with 50 μg/ml erythromycin selection with 5 technical replicates. Plates were incubated at 37 °C overnight (16 h). Colony-forming units (CFU) were calculated for each sample normalized to the weight of tissue input to generate CFU/g tissue. To generate CFU/g stool, one pellet of stool was placed into a 1.5 ml microcentrifuge tube and manually homogenized in PBS with a pipette tip and rigorous pipetting. Serial dilutions were spotted onto an LB agar place with 50 μg/ml erythromycin and incubated at 37 °C overnight. CFU was normalized to weight of the stool.

## Clinical trial design

This study was an interventional, double-blind, dual-centre, prospective clinical trial (WHO Uni-versal Trial Number U1111-1225-7729, ANZCTR number ACTRN12619000210178 https://www.anzctr.org.au/Trial/Registration/TrialReview.aspx?id=376574 registered 13 Feb 2019). See supplementary information for study protocol. The study was approved by the Human Research Ethics Committee of the Central Adelaide Local Health Network (HREC/18/CALHN/751) to meet the requirements of the National Statement on Ethical Conduct in Human Research in accordance with the Declaration of Helsinki for medical research involving human subjects. The study objective was to evaluate the colonization of matched normal and neoplastic bowel tissue by the probiotic E. coli Nissle (EcN). Adult participants undergoing routine colonoscopy or surgical resection for primary colorectal cancer were recruited from St. Andrew's Hospital and Royal Adelaide Hospital, Adelaide ($N = 35$). Written, informed consent was provided before participants were assigned to take either 2 tablets ($10^9$CFU) per day of non-genetically modified EcN (Mutaflor) or placebo for 14 days, prior to their procedure. Patients and treating physicians were blind to active or placebo status. Participants were provided with probiotic tablets and instructed to begin 14 days prior to resection. On the morning of resection investigators verbally confirmed with the participants that they had taken the entire probiotic course, stopping on the day before surgery. There were no alternate methods employed to validate treatment uptake, other than PCR detection of EcN DNA sequence in microbial cultures from tissue samples. Mucosal biopsies (colonoscopy) or surgical resection samples from normal and neoplastic tissue were collected from each participant at the time of their procedure. Participants were excluded if they took probiotics or antibiotics during the trial period. First participant recruited 7 March 2019, last participant 24 September 2019. Eight participants withdrew from the study primarily due to treatment plan change, i.e., no longer undergoing surgery and 2 participants did not have sufficient tumor tissue present to sample from surgery and were excluded. The trial was terminated early after accrual of 35 of the planned 110 participants due to COVID-19-related restrictions and concerns colibactin-expressing E. coli may be pro-carcinogenic. The planned primary outcome of neoplastic colonization status of the probiotic in CRC patient tissues is reported herein, the secondary outcome of microbiome analyses associated with colonization status was discontinued due to smaller than expected sample size.

## Human tissue sample analysis

Initially, participants ($n = 15$) tissue samples were snap-frozen for subsequent DNA extraction. Interim analysis of these samples indicated that detection of EcN was hampered by the presence of host nucleic acid that far outnumbered EcN-derived nucleic acid sequences. These

15 samples were not included in further analyses. To enrich for microbial content in the samples we altered our sample collection methodology as follows. Tissue samples were weighed and collected in sterile 20% glycerol in PBS ($n = 10$ participants). Tissue was immediately homogenized in gentle MACS C Tubes (Miltenyi Biotec, 130-093-237), with a gentle MACS Dissociator (Miltenyi Biotec, 130-093-235), program E. Aliquoted, homogenized tissue was stored at −80 °C until further use. For culture enrichment, the equivalent of 10 mg of human tissue in homogenate was added to 1.2 ml of LB broth/sample and incubated with shaking at 37 °C for 24 h. Culture OD was monitored hourly for the first 14 h to ensure exponential growth, samples from 2 patients were excluded due to inability to attain log phase cultures from tissue homogenates. 1 ml of saturated culture at 24 h was centrifuged at $10,000 \times g$ to collect cells and DNA extracted from cell pellet using DNeasy PowerSoil Pro kit (Qiagen, 47016) for samples from the remaining 8 patients.

### Development of EcN strain-specific PCR assay for human samples

We first tested EcN pMUT2 primers ECN7/8 and 9/10 used previously to detect EcN in mouse fecal samples[47], but found that they generated unacceptable false positives using gDNA isolated from human tissue samples from untreated patients. Alignment of PCR primer sets ECN7/8 or 9/10 against DNA sequences using Primer-BLAST suggested that *Edwardsiella* and *Plesiomonas* contain highly related sequences potentially also found in the human gut, that may cause false positive calls via PCR assay using these primers. To avoid this confounding amplification during EcN detection, we designed a nested PCR strategy to boost specificity and sensitivity for use with human samples using DNA sequence from pMUT2 unique to EcN in comparison with human gut microbiota sequences[66] (Fig. S5). The external 283 bp amplicon spans the unique pMUT2 DNA region: ext-F 5′ CGCGAACGTTAAA-TAATCATC; ext-R 5′ TCTGTTTTAGATAAGGCCATGTCTTC, and was amplified from 50 ng DNA input using KAPA Probe qPCR Master Mix (Roche, KK4716) with PCR conditions: denaturation 95 °C for 20 s; 10 cycle s of 95 °C for 1 s, 60 °C for 20 s, and 72 °C for 25 s. Then 1 μl of this reaction was used as the template for the second 114 bp nested primer/probe-based assay. Nested primer and probe sequences were: int-F 5′ ACCCATCGATAC-CAAATGTATGT; int-R 5′ TCAATGCGTACTCGAC-TATTCAAA; probe 5′ /56-FAM/CCCG-CAGAT/ZEN/CACTGACCTCAA-TACA/3lABkFQ/ using KAPA Probe qPCR Master Mix with PCR conditions as follows: 95 °C for 20 s, 40 cycles of 95 °C for 1 s, 60 °C for 20 s, and 72 °C for 25 s. For 16S PCR, standard KAPA SYBR (non-nested) qPCR Master Mix (Roche, KK4602) with primers reported to amplify a 466 bp amplicon covering 331-797 of the *E. coli* 16S rRNA gene 16S-F 5′ TCCTACGGGAGGCAGCAGT and 16S-R 5′ GGACTACCAGGG-TATCTA ATCCTGTT38. EcN PCR standards were generated from serially diluted DNA isolated from exponentially growing cultures from crushed Mutaflor capsule in LB at 37 °C, with CFU determined by plating of matched samples on LB agar plates. Limit of detection of the assay was based on standard curve dilution series, that is the most dilute standard for which a specific PCR amplicon was reliably generated was determined to be the limit of detection of the assay. This is indicated by a red dashed line in figure, dot points above the line have detectable PCR amplicon signal, those below are beyond the limit of reliable detection of the assay.

### In vitro sample preparation for salicylate metabolite detection

Overnight cultures of EcN-salicylate strain variants were $OD_{600}$ matched to be 1. Cultures were then centrifuged at $3000 \times g$ and 1 mL of the supernatant was collected and stored at −80 °C, the rest was decanted, and the pellet was stored at −80 °C as well until analysis. To 1 ml of supernatants, 1000 μL of extraction solvent ¬(MeOH/MeCN/H$_2$O (2:2:1; v/v/v) containing 0.1 mg/mL of D4-salicylate) was added. Similarly, to cell pellets, 800 μL of the same extraction solvent with the internal D4-salicylate standard was added. Samples were homogenized using a Bead Ruptor 4 at speed 4 for 10 s for 5 cycles. The homogenized samples were centrifuged for 5 min at $14,000 \times g$. Then, the supernatant was removed and dried on the Gene Vac for 5 h and resuspended in 200 μL of MeCN/H$_2$O before LC-MS analysis. A quality control (QC) sample was prepared by combining 20 μL of each sample to assess the reproducibility of the features through the runs.

### Ultra-high performance liquid chromatography (UPLC) analysis

Chromatographic separation was carried out at 40 °C on Acquity UPLC BEH C18 column (2.1 × 50 mm, 1.8 μm) over a 7-min gradient elution. Mobile phase A consisted of water and mobile phase B was acetonitrile both containing 0.1% formic acid. After injection, the gradient was held at 99% mobile phase A for 0.5 min. For the next 4 min, the gradient was ramped in a linear fashion to 50% B and held at this composition for 1 min. The eluent composition returned to the initial condition in 0.1 min, and the column was re-equilibrated for an additional 1 min before the next injection was conducted. The flow rate was set to 450 μL/min and Injection volumes were 2 μL using the flow-through needle mode in the negative ionization mode. The QC sample was injected between the samples and at the end of the run to monitor the performance and the stability of the MS platform.

### Mass spectrometry (MS) analysis

The Synapt G2-Si mass spectrometer (Waters, Manchester, UK) was operated in the negative electrospray ionization (ESI) modes. A capillary voltage of −1.5 kV and a cone voltage of 30 V was used. The source temperature was 120 °C, and desolvation gas flow was set to 850 L/h. Leucine enkephalin was introduced to the lock mass at a concentration of 2 ng/μL (50% ACN containing 0.1% formic acid), and a flow rate of 10 μL/min for mass accuracy and reproducibility. The data was collected in duplicates in the centroid data-independent (MSE) mode over the mass range $m/z$ 50 to 650 Da with an acquisition time of 0.1 s per scan. The QC sample, D4-salicylate, and salicylate standards were also acquired in enhanced data-independent ion mobility (IMS-MSE) in negative modes for the structural assignment. The ESI source settings were the same as described above. The traveling wave velocity was set to 650 m/s and the wave height was 40 V. The helium gas flow in the helium cell region of the ion-mobility spectrometry (IMS) cell was set to 180 mL/min to reduce the internal energy of the ions and minimize fragmentation. Nitrogen as the drift gas was held at a flow rate of 90 mL/min in the IMS cell. The low collision energy was set to 4 eV, and the high collision energy was ramped from 25 to 50 eV in the transfer region of the T-Wave device to induce fragmentation of mobility-separated precursor ions.

### Data pre-processing and statistical analysis for mass spectrometry

All raw data files were converted to netCDF format using DataBridge tool implemented in MassLynx software (Waters, version 4.1). Then, they were subjected to peak-picking, retention time alignment, and grouping using XCMS package (version 3.2.0) in R (version 3.5.1) environment. Technical variations such as noise were assessed and removed from extracted features' list based on the ratios of average relative signal intensities of the blanks to QC samples (blank/QC > 1.5). Also, peaks with variations larger than 30% in QCs were eliminated. The detected signal intensity of salicylate in the samples were normalized to the signal intensity of labeled D4-salicylate. Group differences in measured salicylate levels were calculated using the Welch t-test, $p$-value < 0.05 in GraphPad prism. For Supplementary Fig. 9 targeted analysis was performed as previously described[67]. In brief, samples were separated by liquid chromatography on an Agilent 1290 Infinity LC system by injection of 3 μl of extract through an Agilent InfinityLab Poroshell 120 HILIC-Z, 2.1 × 150 mm, 2.7 μm (Agilent Technologies) column heated to 50 °C. Solvent A (100% water containing 10 mM

ammonium acetate, 5 mM InfinityLab Deactivator Additive and adjusted to pH 9 using ammonium hydroxide) and Solvent B (85% acetonitrile/15% water containing 10 mM ammonium acetate, 5 mM InfinityLab Deactivator Additive and adjusted to pH 9 using ammonium hydroxide) were infused at a flow rate of 0.250 ml min$^{-1}$. The 26-min normal phase gradient was as follows: 0–2 min, 96% B; 5.5–8.5 min, 88% B; 9–14 min, 86% B; 17 min, 82% B; 23–24 min, 65% B; 24.5–26 min, 96% B; followed by a 10-min post-run at 96% B. Acquisition was performed on an Agilent 6230 TOF mass spectrometer (Agilent Technologies) using an Agilent Jet Stream electrospray ionization source (Agilent Technologies) operated at 3500 V Cap and 0 V nozzle voltage in extended dynamic range, negative mode. The following settings were used for acquisition: The sample nebulizer set to 35 psi with sheath gas flow of 12 L min$^{-1}$ at 350 °C. Drying gas was kept at 350 °C at 13 L min$^{-1}$. Fragmentor was set to 90 V, with the skimmer set to 45 V and Octopole Vpp at 750 V. Samples were acquired in centroid mode at 1 spectra/s for $m/z$ values from 50 to 1700.

### Urine sample preparation for salicylate metabolite detection
Urine samples were collected from mice 24 h after EcN-salicylate strain variant oral dosing and frozen at −80 °C for later LC-MS analysis. For LC-MS analysis, urine samples were thawed on ice and polar metabolites were extracted with addition of 100% ice-cold LC-MS grade methanol with 0.1 M formic acid (4:1 ratio of extraction solvent volume to urine). Samples were then vortexed and centrifuged at 20,000 × $g$ for 10 minutes at 4 °C. Supernatants were then transferred to clean LC-MS tubes and loaded onto the LC-MS autosampler, which was temperature-controlled at 4 °C.

### Data acquisition and analysis for mass spectrometry
Raw data was acquired from the instrument and analyzed using previously described open-source XCMS software[68]. Metabolites were identified from ($m/z$, rt) pairs by retention time comparison with authentic standards.

### Histology
For the $Apc^{Min/+}$ mouse model, all intestinal tissue was excised with the cecum removed and tissue was bisected such that there were 5 total sections: duodenum, proximal jejunum, distal jejunum, ileum, and colon. Intestines were flushed with PBS, splayed open, swiss-rolled, and fixed overnight in 4% paraformaldehyde. After 24 h, the Swiss rolls were switched to 70% ethanol and sent for histology services at Histowiz, where they were paraffin-embedded, sectioned, and stained with either H&E or specific immunohistochemistry markers (HA-Tag C29F4 #3724 from Cell Signaling Technology; GranzymeB Leica Biosystems PA0291; CD3 Abcam 16669; CD8 catalog #CST98941 clone D4W2Z). Tumor sizes and IHC quantification were determined using FIJI software image analysis tools.

For the orthotopic CRC transplant model, mice were intraperitoneally injected with Pimonidazole-HCl (Hypoxyprobe, 60 mg/kg) or PBS negative control for Hypoxyprobe staining, 1 h prior to euthanasia. Tumor, nearby adjacent colonic tissue and kidneys were collected at necropsy, rinsed in PBS and fixed overnight in formalin prior to dehydration in 70% ethanol and paraffin-embedding. An additional positive control tumor sample for optimizing ISH staining was generated by intratumoral injection of EcN-lux into a dissected tumor sample from a PBS-treated mouse ex vivo, prior to fixation. Formalin-fixed and paraffin-embedded tissue sections were stained with H&E. Serial sections were also subjected to co-immunofluorescence staining against Hypoxyprobe (cat# HP12-200 Kit, 1:200 dilution) and lipopolysaccharide (LPS, cat# HM6011-100UG, 1:500 dilution) or chromogenic in situ hybridization using RNAscope technology (RNAscope 2.5 Detection Kit, Advanced Cell Diagnostics) following the manufacturer's instructions with a custom probe to detect the lux transcript in EcN-lux or negative control probe, DapB (target region 414-862; catalog number 310043).

Briefly for ISH, tissue sections were baked in a dry oven (HybEZ II Hybridization System, ACD) at 60 °C for 1 h and deparaffinized, followed by incubation with Hydrogen Peroxide (Lot# 322000, ACD) and targeted retrieval (Lot# 322330, ACD). Slides were incubated with relevant probes for 2 h at 40 °C, followed by successive incubations with Amp1 to 6 reagents. Staining was visualized with DAB. For IF studies, sections were treated with blocking buffer (X0909, Dako) for 30 min, incubated with the indicated primary antibodies overnight at 4 °C, and washed with PBS. Sections were then incubated with Alexa Fluor 488/594-conjugated secondary antibodies (1:200 dilution, Thermo Fisher Scientific) for 1 h at room temperature. The sections were then mounted with Vectashield antifade mounting medium (Cat# H-1000-10, Vector Laboratories), and fluorescence was examined using a confocal laser-scanning microscope (FV3000, Olympus).

Resected human CRC samples were fixed overnight in formalin prior to dehydration in 70% ethanol and paraffin-embedding. Formalin-fixed and paraffin-embedded tissue sections were stained with H&E. Serial sections were also subjected to immunofluorescence staining against lipopolysaccharide (LPS, cat# HM6011-100UG, 1:500 dilution). Sections were treated with blocking buffer (X0909, Dako) for 30 min, incubated with the primary antibody overnight at 4 °C, and washed with PBS. Sections were then incubated with Alexa Fluor 488-conjugated secondary antibody (1:200 dilution, Thermo Fisher Scientific) for 1 h at room temperature. The sections were mounted with Vectashield antifade mounting medium (Cat# H-1000-10, Vector Laboratories), and fluorescence was examined using a confocal laser-scanning microscope (FV3000, Olympus).

### Reporting summary
Further information on research design is available in the Nature Portfolio Reporting Summary linked to this article.

## Data availability
The data that support the findings of this study are available within the paper and its Supplementary Information. The full imaging datasets and the de-identified participant data can be shared upon request to the corresponding authors. The clinical trial study protocol is available as Supplementary Note in the Supplementary Information file. Source data are provided with this paper.

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

## Acknowledgements

This work was supported in part by the National Institute of Health U01CA247573 (T.D.), National Institute of Health R01CA24916 (T.D.), the National Science Foundation Graduate Research Fellowship (1644869 to C.R.G.), Pershing Square Foundation (PSF) PSSCRA CU20-0730 (T.D.), Cancer Research Institute (CRI) CRI 3446 (T.D.) National Health and Medical Research Council (APP1184925 to S.L.W.), Cancer Council SA Beat Cancer Project on behalf of its donors and the State Government of South Australia through the Department of Health (MCF0418 to S.L.W.), the Gastroenterological Society of Australia Bushell Post-doctoral Research Fellowship (S.L.W.), the Faculty of Health Science at the University of Adelaide (S.L.W.), the South Australian Health and Medical Research Institute (S.L.W.).

## Author contributions

C.R.G. and C.C. performed the in vivo colonization studies in the ApcMin/+ model, G.A.R., E.T., T.R.M.L. and T.W. in the orthotopic CRC models. C.R.G. analyzed the ApcMin/+ histology images with input and guidance from S.R.T. H.K. provided pathology analysis of orthotopic CRC model. C.R.G., S.R.T. and G.A.R. performed the IHC and ISH staining not performed by Histowiz. J.I., YU.J. and T.C., designed and cloned the salicylate-producing strain. C.R.G., J.I. and YU.J. designed and/or engineered the EcNΔclbA strains and salicylate-producing strains. C.R.G. performed all ex vivo assays in the ApcMin/+ model with assistance from I.L. and C.C. G.A.R. performed all ex vivo assays in the orthotopic CRC model with assistance from L.V., J.A.W. and E.T. F.Z. performed all cell-culture-based mass spectrometry and A.A.S. and K.R. performed all urine-based mass spectrometry. T.W., E.T. and J.Q.N. monitored animals for orthotopic CRC model. L.V., S.L.W., T.W., M.La. T.S. M.T., M.Le., L.P., J.P., T.F., P.K., A.L., A.M.S., D.A.G. and D.L.W. recruited participants, collected clinical data, and analyzed samples from clinical trial. J.M., C.O., G.A.R., G.R., M.La., T.S., M.T., M.Le., L.P., J.P., T.F., P.K., A.L., N.A., T.D., D.L.W., C.R.G. and S.L.W. provided intellectual input into study design. C.R.G., A.S., T.D., S.L.W. and D.L.W. wrote the manuscript with input from all authors.

## Competing interests

T.D., N.A., D.L.W., S.L.W. and C.R.G. have financial interest in GenCirq Inc. T.D., D.L.W., S.L.W. and C.R.G. have filed a provisional patent application ("Colorectal Cancer Screening, Prevention, And Treatment With Engineered Probiotics") with the US Patent and Trademark Office related to this manuscript. The remaining authors have no other competing interests.

## Additional information

Candice R. Gurbatri[1,15], Georgette A. Radford[2,15], Laura Vrbanac[2], Jongwon Im [1], Elaine M. Thomas[2], Courtney Coker [1], Samuel R. Taylor [3], YoungUk Jang[1], Ayelet Sivan[1], Kyu Rhee [4], Anas A. Saleh[4], Tiffany Chien[1], Fereshteh Zandkarimi [5], Ioana Lia[1], Tamsin R. M. Lannagan[2], Tongtong Wang [2,6], Josephine A. Wright[6], Hiroki Kobayashi [2,6], Jia Q. Ng[2], Matt Lawrence[7], Tarik Sammour[2,6,7], Michelle Thomas[7], Mark Lewis[7], Lito Papanicolas[6,8], Joanne Perry[7], Tracy Fitzsimmons[7], Patricia Kaazan[2], Amanda Lim [2], Alexandra M. Stavropoulos[2], Dion A. Gouskos[2], Julie Marker[9], Cheri Ostroff [10], Geraint Rogers[6,8], Nicholas Arpaia [11,12], Daniel L. Worthley [6,13], Susan L. Woods [2,6,16] ✉ & Tal Danino [1,12,14,16] ✉

[1]Department of Biomedical Engineering, Columbia University, New York, NY 10027, USA. [2]Adelaide Medical School, University of Adelaide, Adelaide, SA 5000, Australia. [3]Weill Cornell-Rockefeller-Sloan Kettering Tri-Institutional MD-PhD program, New York, NY, USA. [4]Division of Infectious Diseases, Weill Department of Medicine, Weill Cornell Medicine, New York, NY, USA. [5]Department of Chemistry, Columbia University, New York, NY, USA. [6]South Australian Health and Medical Research Institute (SAHMRI), Adelaide, SA 5000, Australia. [7]Colorectal Unit, Department of Surgery, Royal Adelaide Hospital, Adelaide, SA 5000, Australia. [8]College of Medicine and Public Health, Flinders University, Bedford Park, South Australia 5042, Australia. [9]Cancer Voices SA, Adelaide, South Australia, Australia. [10]University of South Australia, Adelaide, South Australia 5000, Australia. [11]Department of Microbiology & Immunology, Vagelos College of Physicians and Surgeons of Columbia University, New York, NY 10032, USA. [12]Herbert Irving Comprehensive Cancer Center, Columbia University, New York, NY 10027, USA. [13]Colonoscopy Clinic, Spring Hill 4000 Queensland, Australia. [14]Data Science Institute, Columbia University, New York, NY 10027, USA. [15]These authors contributed equally: Candice R. Gurbatri, Georgette A. Radford. [16]These authors jointly supervised this work: Susan L. Woods, Tal Danino. ✉e-mail: susan.woods@adelaide.edu.au; td2506@columbia.edu

