## [Peer Review File · Nature Communications]

Engineering tumor-colonizing E.coli Nissle 1917 for detection and treatment of colo-rectal neoplasiaREVIEWER COMMENTS

Reviewer #1 (Remarks to the Author): with expertise in bacteria-based cancer therapy, bacteria engineering

In this work, Candice R. Gurbatri and the co-author developed bioengineered probiotics which makes the screening and treatment of colorectal cancer (CRC) possible. First, authors found the phenomenon of selective, long-term colonization of probiotic *E. coli* Nissle 1917 (EcN) in colorectal adenomas after orally administration. Then they detected lesions by EcN-luc from stool or engineered EcN producing a small molecule-salicylate from the urine. Finally, they also engineered EcN to locally release immunotherapeutics at the neoplastic site. It is an interesting work. However, I will not be convinced that the engineered EcN to be a screening and therapeutic platform unless the following issues were solved and clarified. The overall quality of the article needs to be improved.

1. Wild-type EcN-luc showed retention in the *ApcMin/+* mouse gut for up to 7 weeks after oral administration. However, EcN Δ clbA were only detected for one week in SI 2. Besides, in figure legend of SI2, (C) one stool pellet was collected 24, 48, and 72h after last dose, homogenized, plated on antibiotic-selective plates and quantified for CFU (n=3, n=1 stool per mouse). (B) After 1 week, mice were sacrificed, intestinal tissue was excised and ex vivo imaged for bioluminescence. Red arrows point to macroadenomas on distal intestinal tissue (representative image from sample size of n=5 mice). It seems the marker of B and C was wrong.

2. *ApcMin/+* mice develop multiple intestinal adenomas. Then abundant of EcN-luc and EcN-SA can colonize in many tumors. It may be helpful for the detection. What about the organoid orthotopic model which only developed one or two tumors? Besides, in figure 3C, what is the representation of green dots and black dots?

3. In Figure S8C, an increased percentage of smaller adenomas was observed in SLIC-3 treated mice. Then how to judge its effectiveness? Perhaps there is no difference in tumor number between different groups.

4. Have you tried oral SLIC and injected with granulocyte-macrophage colony-stimulating factor (GM-CSF), and blocking nanobodies against PD-L1 and CTLA-4 targets as control group?

5. Please evaluate the tumor CD3+, CD8+ cells by FCM flow cytometry instead of immunohistochemical (IHC) techniques.

6. In the experiment of SLIC3 producing immunotherapeutics, the therapy efficiency and changes of tumor-immune microenvironment is limited. The author can discuss it in the discussion. And it is recommended to try in another orthotopic model.

7. It should be better to present the timeline scheme of the detection and treatment experiment. Please provide the details process of engineering different EcN in materials and methods.

8. There are issues in statistical line between different groups, e.g., Figure 4 c (PJ, I), e, f. It is not friendly for readers.

9. The author mentioned in the article, "Taken together, these data lay the groundwork for future pre-clinical and clinical experiments testing engineered EcN for CRC screening, prevention and treatment". Is it accurate to use prevention based on these data?

Reviewer #2 (Remarks to the Author): with expertise in colorectal cancer

The authors report their findings to suggest the potential benefits of bioengineered-EcN bacteria colonizing colorectal tumours in an ApcMin^{+/-} mouse model. This manuscript needs significant polishing before publication.

-Title needs updating since it does not convey this study's important message or findings.

- Introduction should cover essential topics to understand a study's relevance in the research field. Although EcN is a well-known tumour-colonizing bacteria, the authors did not cite previous reports. Moreover, salicylate and its large-medical history have yet to be mentioned. How some previous research groups have bioengineered that bacteria to synthesize it is unclear too. Please, update this section.

-Aims need clarification. Since the introduction needs more complexity, the current aims are

a summary of key findings rather than specific goals to fill in the gaps of previous reports.

Overall, the results section lacks the complexity seen in NC's studies.

Results need sub-sections and headings summarizing the essential findings of each one. The rationale needs to be included between different experimental sets.

Please, provide the median diameter of tumours per mouse in both PBS-treated and EcN-lux colonized groups.

Please, clarify the relevance of comparing the median tumour diameter between mice and human patients. Ref 34 (Am J Gastroenterol. 2020 Mar;115(3):435-464) is a practical guideline and does not fit into this discussion. Please, clarify.

Please, demonstrate that EcN co-localizes with mucin lakes or mucin-producing cells in mouse and human tumours.

The authors should experimentally demonstrate which chemical signalling attracts EcNs to mucin lakes or mucin-producing EcN in their model.

Even though APC genetic silencing is the driver mutation leading tumours to develop in both GEM mice and humans, someone should also notice that Apc mutation is not the natural causation of spontaneous murine colorectal tumours (a rare event in those mammals), ApcMin^{+/-} mice do not mimic the microbiota composition found in human colorectal tumours. For instance, intestinal tumours instead of colorectal lesions develop in ApcMin^{+/-} mice. Many studies addressed this issue, demonstrating that intestinal and colonic microbiota have distinct compositions. A growing body of evidence has also proposed more accurate murine colorectal cancer models, which the current study's authors did not take advantage of. Unfortunately, this study has a fundamental modelling flaw. To complement their preliminary but exciting findings, the authors should address these concerns by developing experiments in current GEM models mimicking the human colorectal tumoural condition and microbiota composition.

It is unclear why the authors used antibiotic treatment instead of gnotobiotic mouse models. It needs updating.

The dataset shown in Figure 3 requires a larger N per group of independent experiments. Reproducibility is questionable.

The description related to Fig 4A, S4, and S6 needs to be clearly shown. The authors hint at potential outcomes from these experiments but need to exhibit the findings from these experiments. Please, give the numbers to the readers.

The dataset shown in Fig 4B and 4C requires a larger N per group of independent experiments. The authors should display the number of tumours per mouse instead of their numbers per cm². Reproducibility is questionable.

Fig 4E, 4F, 4G - the authors should show % of positive cells per mouse instead of them per polyp. The statistical analysis strategy needs to be revised. Please, update these graphs to match the number of mice in each group.

Fig 4E, 4F - the authors should perform an in-situ flow cytometry analysis to characterize their CD8 population properly. These results are questionable as they are now. Fig 4G needs significance to the proposed mechanism.

Supplementary figures. Overall, they need updating and grouping following their citing order in the manuscript. There is no need for single images to be shown as figures when the latter is expected to be built as highly complex panels.

Reviewer #3 (Remarks to the Author): with expertise in synthetic biology

Revision nature comms.

In this study, the authors demonstrate that probiotic EcN can be used in mice to monitor for intestinal lesions such as adenoma. Moreover, by using EcN as a frame, the authors can release several therapeutic molecules in the intestine, which greatly reduces the size of tumors.

Although the results of this work are relevant to the field, several aspects must be taken into consideration:

The authors claim that "This phenomenon of neoplasia selective colonization of EcN suggested its utility as a platform to both monitor adenoma presence and also then to manipulate lesions in situ." Colonization of lesions and modulation of inflammation with EcN have already been reported in hepatic lesions (doi.org/10.1080/19490976.2021.1924261). In this study, the authors show that this phenomenon of colonizing lesions and modulating inflammation also happens in the intestine.

- Are there known mechanisms that enable EcN to colonize intestinal lesions and not

colonize a healthy intestine? Authors should address this issue in the manuscript.

- The use of EcN expressing luxCDABE allows monitoring the presence of intestinal lesions such as adenomas thanks to the ability of EcN to colonize these lesions. However, the reported data do not show that colonization of intestinal lesions is restricted to adenomas or carcinomas. What are the limits of this method for identifying adenomas and carcinomas and distinguishing them from other types of intestinal lesions that are not linked to colon cancer? Authors should discuss in the text the limitations of this approach for discrimination between different types of lesions.

- For this reviewer, the benefits of detecting EcN by detecting salicylate in urine relative to detecting EcN in fecal samples are not clear. Although it is true that there are easy-to-use commercial kits for detecting salicylate in the urine, there are a number of problems with the use of salicylate to identify the presence of EcN in the intestine after having administered the probiotic. In a possible application in humans maybe salicylate is not the best option. Levels of salicylate in the urine can vary considerably from person to person depending on aspects such as diet or blood pH, among others. This may limit the use of salicylate as an indicator of colonization of intestinal lesions by EcN. Fecal samples appear to be far more reliable by counting the CFU/ml present. The authors should highlight the limits that the use of salicylate as a detection molecule could have for future applications in uncontrolled systems (for example in human applications).

Why do the authors decide to use salicylate?

- Regarding the proposed immunotherapy, based on the release of nanobodies blocking PD-L1 and CTLA-4 targets and cytokines like GM-CSF there are several aspects to discuss:

- a. Why was it chosen to induce cell lysis to release therapeutic molecules into the extracellular environment instead of using one of the endogenous secretory pathways in EcN?

Cell lysis induced by the expression of a lysis gene (phiX174E) controlled by a circuit as shown in Figure 4a implies a very significant reduction of the colonizing EcN population

when AHL levels exceed a certain critical threshold. This is contradictory to one of the aspects that the authors highlight as positive of the use of EcN, i.e. the ability to colonize the lesions and remain in them for long periods of time. The authors must justify the choice of this strategy despite the reduction in EcN colonization.

b. The circuit in Figure 4a is a circuit based on quorum sensing. The figure is incomplete because this circuit requires expression of the LuxR receptor protein (authors should indicate the production of this protein in Figure 4a). These quorum sensing based circuits exhibit a very ON/OFF response. When the bacterial population exceeds some critical density, the expression of the lysis gene increases dramatically. The critical density required to activate the lysis circuit depends not only on the density of EcN but also, and very decisively, on the expression levels of the LuxR protein (DOI: 10.1093/nar/gku964). Different levels of LuxR expression may imply changes in orders of magnitude in lactone levels (AHL) and hence in the EcN concentration required to activate lysis gene expression.

I wonder how active is the lysis circuit according to the LuxR expression levels in the particular circuit used?

Authors must provide information on the strength of the promoter and RBS under which LuxR is expressed. It is also necessary to know the strength of the RBS upstream the phiX174E gene is expressed.

An in vitro characterization of this circuit (transfer function) is necessary to show the relationship between the density of EcN and the expression levels of phiX174E.

- How the authors can be sure that these levels of concentration of EcN in colonized areas are enough to be sure that the lysis of EcN takes place significantly in the intestine?

This could have been easily monitored by determining the quantity of CFU/ml present in fecal samples. A significant reduction in the presence of EcN in faecal samples would be a clear indication that EcN lysis is occurring in the gut.

Based on the data provided by the authors, it is not possible to determine the effectiveness of the applied lysis mechanism.

- According to the above arguments, surprisingly, the reported data show that

improvements in mice with EcN-SLIC are close to those achieved with SLIC-3 (Figure S8). However, assuming the release of therapeutic molecules after EcN lysis, one would expect the impact to be much more significant. This would be compatible with the fact that only a partial lysis of the EcN population may take place. Probably because the densities of EcN are insufficient to fully activate the lysis circuit. These findings suggest that only the probiotic effect of EcN can be sufficient to achieve a notable improvement in terms of tumor area reduction. Of course, the eventual partial release of therapeutic molecules enhances the results. Why do the authors not address in the text the improvements achieved with EcN-SLIC and concentrate only on the results of EcN-SLIC3?

I wonder if these improvements would not be even greater if the expression of the lysis gene were done in an inducible system (for instance induced by molecule that can be supplied to mice) and not depending on cell density.

- Finally, one last minor question is what antibiotic authors use to select EcN on LB plates. Is it resistance that has been introduced in the EcN strain or is it natural EcN resistance to antibiotics like erythromycin or vancomycin?

Summary of point-by-point responses:

We would like to thank the reviewers for their feedback that led to substantial improvement of the manuscript. Briefly, in this revised version, we have included several suggested experiments including incorporation of additional mouse models to highlight the robustness of probiotic colonization. We have also included data from a small clinical trial study with corresponding microscopy images to visualize bacteria in human colorectal tissues. Finally, we have included more experimentation demonstrating significantly improved salicylate production and corresponding in vivo readout.

Reviewer #1 (Remarks to the Author): with expertise in bacteria-based cancer therapy, bacteria engineering

In this work, Candice R. Gurbatri and the co-author developed bioengineered probiotics which makes the screening and treatment of colorectal cancer (CRC) possible. First, authors found the phenomenon of selective, long-term colonization of probiotic E. coli Nissle 1917 (EcN) in colorectal adenomas after orally administration. Then they detected lesions by EcN-luc from stool or engineered EcN producing a small molecule-salicylate from the urine. Finally, they also engineered EcN to locally release immunotherapeutics at the neoplastic site. It is an interesting work. However, I will not be convinced that the engineered EcN to be a screening and therapeutic platform unless the following issues were solved and clarified. The overall quality of the article needs to be improved.

We thank Reviewer 1 for their interest in our manuscript and appreciate the opportunity to improve upon its quality.

1. Wild-type EcN-lux showed retention in the ApcMin/+ mouse gut for up to 7 weeks after oral administration. However, EcN Δ clbA were only detected for one week in SI 2. Besides, in figure legend of SI2, (C) one stool pellet was collected 24, 48, and 72h after last dose, homogenized, plated on antibiotic-selective plates and quantified for CFU (n=3, n=1 stool per mouse). (B) After 1 week, mice were sacrificed, intestinal tissue was excised and ex vivo imaged for bioluminescence. Red arrows point to macroadenomas on distal intestinal tissue (representative image from sample size of n=5 mice). It seems the marker of B and C was wrong.

These captions have been corrected in the revised manuscript. Regarding duration of colonization of the EcN Δ clbA, we have demonstrated colonization up to 1 week. As we undertake all our animal experiments by our animal ethics principles of 'Replace, Refine, Reduce' to minimize the number of animals needed to generate equivalent datasets (in this case duration of colonization by WT EcN in comparison to EcN Δ clbA), we have prioritized these mice for other experiments that require earlier sacrifice timepoints and so haven't taken EcN Δ clbA colonization experiments out longer. This is because previously published results by Kalantari et al PLOS One 2023, demonstrate that disruption of Δ pks does not confer a competitive advantage or disadvantage for the Δ pks strain with regards to gut transit time or colonization. We have also undertaken further fitness testing of the WT and EcN Δ clbA strains and consistently observe similar behavior of the two strains. We have included this in **Revised Fig. 1F** in the manuscript.

2. ApcMin/+ mice develop multiple intestinal adenomas. Then abundant of EcN-luc and EcN-SA can colonize in many tumors. It may be helpful for the detection. What about the organoid orthotopic model which only developed one or two tumors? Besides, in figure 3C, what is the representation of green dots and black dots?

We have further explored colonization of EcN-lux in an additional immunocompetent CRC model with discrete and limited tumor burden and have added this data to the manuscript (**Revised Fig. 2**). Once again we demonstrate specific colonization of single tumors using *ex vivo* tumor bioluminescence imaging and CFU plating of dissociated tissues. In this model, we were able to confirm colonization through significantly increased recovery of EcN-lux from stool 1-5 days post gavage, suggesting EcN-lux is detectable in mouse models bearing a single tumor. Regarding Figure 3C, our apologies the green dots depicted the EcN-SA strain and the black dots depicted the EcN-lux strain. We have now replaced this figure with our more extensive set of salicylate producing strains (Figure 3D) and have ensured correct labelling and appropriate legend.

3. In Figure S8C, an increased percentage of smaller adenomas was observed in SLIC-3 treated mice. Then how to judge its effectiveness? Perhaps there is no difference in tumor number between different groups.

We thank the Reviewer for the opportunity to clarify our results regarding Fig. S8C. We demonstrate a decrease in both tumor count (Fig S8B) and total tumor area (Fig 4B) in SLIC-3 treated APCmin/+ mice. This has been further highlighted in the revised manuscript for improved clarity.

Lines 242-244: *Histological analysis of hematoxylin and eosin-stained tumors demonstrate an overall reduction of adenoma area (Fig. 4B) and number (Fig. S8A-B) by ~47% with SLIC-3 treatment.*

4. Have you tried oral SLIC and injected with granulocyte-macrophage colony-stimulating factor (GM-CSF), and blocking nanobodies against PD-L1 and CTLA-4 targets as control group?

We thank the Reviewer for the opportunity to clarify our choice of controls. We have evaluated purified nanobody controls and antibody controls against PD-L1 and CTLA-4 targets in subcutaneous models in previous publications (Gurbatri et al Science Translational Medicine 2020), and concluded that live bacteria producing these therapeutic agents were needed for maximal therapeutic efficacy. Similar results were observed with separate therapeutic targets in Chowdhury et al (Nature Medicine 2019). Given this existing data and working in all our animal experiments by our animal ethics principles of 'Reduce, Replace, Refine' to minimize the number of animals needed to generate equivalent datasets, we proceeded only with the oral treatment of all three bacterial strains in these cohorts and used +/- (empty) SLIC as our control.

5. Please evaluate the tumor CD3+, CD8+ cells by FCM flow cytometry instead of immunohistochemical (IHC) techniques.

While flow cytometry would be more readily quantitative, separation of diseased and healthy tissue is difficult at the macroscopic level in the *APCMin*+/+ model where adenoma presence is variable and present throughout the intestine. Furthermore, tissue dissection and digestion for flow cytometry results in a loss of spatial information that is retained using immunohistochemical methods. We believe that the spatial distribution of the immune cells is the significant result versus absolute quantification as the field aims to induce effective infiltration of anti-tumor immune cells to overcome immune-exclusion (i.e. ineffectual immune cells on the tumor periphery).

6. In the experiment of SLIC3 producing immunotherapeutics, the therapy efficiency and changes of tumor-immune microenvironment is limited. The author can discuss it in the discussion. And it is recommended to try in another orthotopic model.

We have added additional text to acknowledge the limited therapeutic efficacy of SLIC-3 and highlight platform modularity (see below). EcN can be engineered to deliver multiple kinds of therapeutics including toxins that may be more efficient in altering the tumor microenvironment. We have instead demonstrated colonization of EcN in multiple models (**Revised Fig. 2**), highlighting its potential as a therapeutic vector across multiple disease subtypes.

Lines 268-270: Finally, therapeutic efficacy of SLIC-3 is limited, but as our probiotic platform is modular, there is the possibility to expand both screening and therapeutic cargo to explore other therapeutic combinations and achieve enhanced adenoma reduction.

7. It should be better to present the timeline scheme of the detection and treatment experiment. Please provide the details process of engineering different EcN in materials and methods.

More detailed methodology has been included in the revised manuscript. Timelines of detection and treatment experiments are included in **Revised Fig. 3** and **Revised Fig. 4**.

8. There are issues in statistical line between different groups, e.g., Figure4 c (PJ, I), e, f. It is not friendly for readers.

The statistical lines have been adjusted in revised manuscript for improved readability.

9. The author mentioned in the article,” Taken together, these data lay the groundwork for future pre-clinical and clinical experiments testing engineered EcN for CRC screening, prevention and treatment”. Is it accurate to use prevention based on these data?

Our results demonstrate colonization of intestinal precancerous lesions in the *APCMin*^{+/-} model, suggesting the use of the platform for early detection of benign polyps and therefore prevention of CRC. However, as this may be unclear to readers, this sentence has been changed in the revised manuscript (see below) to reflect the use of engineered EcN for CRC early intervention through improved detection and treatment.

Line 288-290: Taken together, these data lay the groundwork for future pre-clinical and clinical testing of engineered EcN for early CRC detection and treatment.

Reviewer #2 (Remarks to the Author): with expertise in colorectal cancer

1. The authors report their findings to suggest the potential benefits of bioengineered-EcN bacteria colonizing colorectal tumours in an ApcMin^{+/-} mouse model. This manuscript needs significant polishing before publication.

-Title needs updating since it does not convey this study's important message or findings.

We have changed the title in the revised manuscript to “Tumor-colonizing E.coli Nissle 1917 for the detection and treatment of colorectal neoplasia”

- Introduction should cover essential topics to understand a study's relevance in the research field. Although EcN is a well-known tumour-colonizing bacteria, the authors did not cite previous reports. Moreover, salicylate and its large-medical history have yet to be mentioned. How some previous research groups have bioengineered that bacteria to synthesize it is unclear too. Please, update this section.

We have included text changes (see below) and additional citations to cover aspects of tumor-colonizing bacteria, the use of EcN in bacterial cancer therapy, and the use of salicylate in CRC. Specifically, citations Chien et al, ACS (2020) and Qi et al, Frontiers in Microbiology (2018) provide methodology to engineer bacteria for salicylate production.

Lines 82-86: Furthermore, genetic conditions that can predispose patients to CRC, such as familial adenomatous polyposis (FAP), result in hundreds of colonic adenomas, the primary precursor lesions of CRC, complicating CRC prevention. While long-term use of non-steroidal anti-inflammatory medications such as aspirin (acetyl salicylic acid) can significantly reduce CRC incidence, protection from conversion to CRC is incomplete.

-Aims need clarification. Since the introduction needs more complexity, the current aims are a summary of key findings rather than specific goals to fill in the gaps of previous reports. The aims have been reframed in the revised manuscript (see below).

Lines 93-100: Here, we assess whether orally delivered E. coli Nissle 1917 (EcN) can selectively colonize adenomas and isolated neoplastic lesions in orthotopic, genetic and transplant murine models of CRC. EcN is a probiotic strain with demonstrated safety and has been investigated as a chassis for other types of cancers. We investigate the capacity for these bacteria to remain colonized long-term, and the utility of EcN for adenoma detection by recovering EcN from stool or by engineering EcN to produce a small molecule measurable in the urine. Lastly, we test whether engineered EcN can deliver immunotherapeutics within adenomas, to manipulate the TME in situ and impact overall disease burden.

2. Overall, the results section lacks the complexity seen in NC's studies.

In the revised manuscript, we now include colonization studies in the *APC^{Min}+* mouse model of CRC predisposition and two orthotopic mouse CRC models representing MSS and MSI subtype disease that resemble human CRC (Lannagan et al, Gut 2019). We have also further optimized our salicylate producing EcN strains to increase the amount of metabolite generated for urine screening approaches. To further address this point we now also move beyond preclinical mouse models to include our clinical trial data (**Revised Fig. 2J-K**), paving the way for future development of our platform in humans.

Results need sub-sections and headings summarizing the essential findings of each one. We have now included subheadings in the revised manuscript.

The rationale needs to be included between different experimental sets.

Please, provide the median diameter of tumours per mouse in both PBS-treated and EcN-lux colonized groups.

We thank the Reviewer for highlighting that further clarification was required here.

The median tumor diameter of the PBS-treated group (n = 4) was 4+/-1.3mm and therefore was larger than the median tumor diameter of the EcN-lux treated group, 2+/-1.2mm (measurements from H&E stained histological sections of each tumor, this information is now provided in the revised manuscript). This reinforces that we had no problems with aseptic technique and confinement protocols across our mouse cages between the PBS and EcN-lux treated animals (which would be detected by the presence of contaminant lux+ bacteria in the PBS group) and that the reason the PBS animals were negative for EcN-lux was because our process was clean, rather than the tumors being smaller in the PBS-treated group. We understand this may have been the reviewer's concern. To provide further clarity regarding the tumor grades in the orthotopically injected cohorts we have also expanded figure S4 and made clear that we only treated mice once tumors had reached grade 3-4 as judged by colonoscopy.

Please, clarify the relevance of comparing the median tumour diameter between mice and human patients. Ref 34 (Am J Gastroenterol. 2020 Mar;115(3):435-464) is a practical guideline and does not fit into this discussion. Please, clarify.

The clinical guidelines referred to include definition of the sizing of 'diminutive' polyps in humans at 0-5mm, hence our inclusion of the reference to support the preceding statement about size of 'diminutive' polyps. The purpose was to give the reader some idea of the size of the lesions we are detecting in our mouse models and how it may relate to target lesions for detection in humans-notwithstanding the fairly obvious differences in overall body size between mice and humans. We have altered the wording to make this clearer.

Lines 151-153: The median diameter of EcN-lux colonized tumors was 2mm (+/-1.2mm), suggesting the size of neoplastic lesions detected using this EcN-lux platform was similar to the size of diminutive (0–5mm) polyps in humans.

3. Please, demonstrate that EcN co-localizes with mucin lakes or mucin-producing cells in mouse and human tumours. The authors should experimentally demonstrate which chemical signalling attracts EcNs to mucin lakes or mucin-producing EcN in their model.

Apologies, thank you for pointing out that this may distract the reader from our key findings. In prior work we generated the MSS orthotopic model of CRC used here (Fig. 2A) and were fascinated to note that inclusion of genetic alterations found in the serrated subtype of CRC resulted in a model that recapitulates features of the human disease, including the distinctive mucinous differentiation phenotype of the tumors (Lannagan et al., Gut 2019). We were again emphasizing the validity of this mouse model as representative of the serrated subtype of CRC with our images of mucinous differentiation of the mouse tumors and matching human serrated subtype tumors (original Fig. 2E). This information is already published and so we have now removed these images from the **Revised Fig. 2** because they are slightly off-point for this work, but leave reference to our prior study describing the model.

As requested, we have also undertaken matched mucin (Alcian Blue and Periodic Acid Schiff, AB-PAS) and EcN-lux staining of tumors from the MSS CRC mice that were administered EcN. We

have also performed mucin staining of human CRC samples with paired localization of bacteria using the pan-gram negative bacterial LPS stain. We have included this data here for the reviewer's information and could include it in the manuscript if requested to do so. However we would prefer not to focus on the location of orally administered EcN or endogenous LPS+ bacteria with respect to mucin, or mucin producing cells. Even though we observe evidence of these being present near EcN in our mouse model, the vast majority of mucin and mucin producing cells in the tumor are distant from EcN locations, i.e. the presence of mucin does not guarantee generation and colonization of an EcN niche. Equally the *APC^{Min}+/+* mouse model is not characterized by polyps displaying mucinous differentiation, yet is still well colonized by EcN. As such, we agree that careful elucidation of key molecular drivers of neoplastic colonization by EcN are essential to further develop this approach and acknowledge this with additional text in the discussion (*Lines 285-289 'while mechanisms such as selective adhesion to tumor antigens and attraction to small molecules produced in tumors that drive translocation of bacterial pathogens across the gastrointestinal barrier, are known for other E coli strains, Salmonella typhimurium and Fusobacterium nucleatum, a deeper understanding of characteristics governing EcN establishment within tumors is needed.'*) This mechanistic characterization is a major important undertaking and falls outside the scope of the current study which focuses on our initial demonstration of the use of orally administered, engineered bacteria for the detection and prevention of colorectal neoplasia *in situ*.

Mucin staining of MSS mouse tumour and human CRC. Orthotopic mouse tumor (shown in Fig. 2E) with serial sections stained for (A) H&E, (B) Alcian Blue (AB) and Periodic Acid Schiff (PAS), (C-D) EcN-lux specific location by RNAscope in situ hybridization for lux at 30X and 10X magnification, respectively. Human CRC (shown in Fig. S4) serial sections for region one stained for (E) H&E and (F) AB and PAS, (G-H) immunofluorescent staining for lipopolysaccharide (LPS, green). Scale bars: 50 μ m (A-C & E-G) or 10 μ m (D & H).

4. Even though APC genetic silencing is the driver mutation leading tumours to develop in both GEM mice and humans, someone should also notice that Apc mutation is not the natural causation of expontaneous murine colorectal tumours (a rare event in those mammals), ApcMin+/- mice do not mimic the microbiota composition found in human colorectal tumours. For instance, intestinal tumours instead of colorectal lesions develop in ApcMin+/- mice. Many studies addressed this issue, demonstrating that intestinal and colonic microbiota have distinct compositions. A growing body of evidence has also proposed more accurate murine colorectal cancer models, which the current study's authors did not take advantage of. Unfortunately, this study has a fundamental modelling flaw. To complement their preliminary but exciting findings, the authors should address these concerns by developing experiments in current GEM models mimicking the human colorectal tumoural condition and microbiota composition.

The *ApcMin/+* model is a commonly utilized, informative mouse model of CRC predisposition (Yang et al Gastroenterology 2022; Tang et al Nature Communications 2023; Zagato et al Nature Microbiology 2020) albeit with the caveat (as noted by the reviewer) that the polyps form in the small intestine rather than colon, as would occur in humans. To address this, we now also include two additional orthotopic mouse CRC models representing MSS and MSI subtype disease that were generated by inclusion of genetic alterations found most commonly in human CRC and that result in discrete tumors in the distal colon that histopathologically resemble human CRC (Lannagan et al, Gut 2019). These tumors in the distal colon are also specifically colonized by EcN (**Fig. 2**). To further address this point we now also move beyond preclinical mouse models to include our clinical trial data (**Revised Fig. 2J-K**). While this promising data from humans requires follow-up in a larger patient cohort, these results suggest the enrichment of EcN in neoplastic tissue is likely to also occur in humans following oral administration of EcN (trade name Mutaflor). As the first report of EcN localization in neoplastic tissue in humans we feel the relatively small trial provides useful information and cohort size is comparable to other early observations published in Nature Communications and similar prestigious translational journals (Mota et al Nature Communications 2022; Gassart et al Science Translational Medicine 2021; Meng et al Science Translational Medicine 2021; Song et al Science Translational Medicine 2023; Roberts et al Science Translational Medicine 2014).

5. It is unclear why the authors used antibiotic treatment instead of gnotobiotic mouse models. It needs updating.

We thank the Reviewer for this suggestion. Research demonstrates that gut dysbiosis is strongly linked to colorectal cancer formation and progression and therefore it was important for us to recapitulate this in our model. However, we are aware this presents complications with the feasibility of translating our work and therefore to address this we have moved forward into models without antibiotic pretreatment, please refer to **Revised Fig. 2G-I**.

6. The dataset shown in Figure 3 requires a larger N per group of independent experiments. Reproducibility is questionable.

We understand the Reviewer's concern for reproducibility and have included data from a separate trial in the revised manuscript of salicylate levels that have been collected in a distinct cohort and

mass spectrometry experiment. The revised data contains an additional 48h time point. Due to differences in experimental design and mass spectrometry runs, this data could not be combined with previous cohorts, but enhanced salicylate levels are still identified in tumor-bearing mice. For reference, a separate experiment included in the first submission is shown again below. Data was collected at a 24h time point and shows significantly elevated salicylate levels in tumor-bearing mice compared to healthy mice. Moreover in earlier studies, the main metabolite of salicylate, salicylic acid (SU) was explored and this data is shown below as well, with tumor-bearing mice having increased SU levels.

Metabolite detection in distinct APC^{Min/+} cohorts. In two separate cohorts, 15-week-old APC^{Min/+} mice were dosed with 10⁹ EcN-SA bacteria and urine was collected 24hr after dosing. (A) LC-MS quantification of salicylate and (B) salicylic molecules in urine of wild-type (WT) and APC^{Min/+} mice (p<0.05, unpaired T test, n=3-5 mice per group).

7. The description related to Fig 4A, S4, and S6 needs to be clearly shown. The authors hint at potential outcomes from these experiments but need to exhibit the findings from these experiments. Please, give the numbers to the readers.

Our apologies for the confusion. We have now included experimental schemata for Fig.2 (**Revised Fig 2A - this is the schema for Fig. 2 and Fig. S6**) and Fig. 4 (**Fig. S8A**) and provided more details and additional images for Fig. S4 to make the experimental details clear for the reader. Animal numbers in each group are provided in the figure legends (**Fig.1 B,D,G,I; Fig.2 C,D,H,I; Fig. 3B,F-G; Fig. 4 B-C,E-G**).

8. The dataset shown in Fig 4B and 4C requires a larger N per group of independent experiments. The authors should display the number of tumours per mouse instead of their numbers per cm2. Reproducibility is questionable.

We thank the Reviewer for this suggestion, but our data aggregates mice across both sexes and 3 independent cohorts. Furthermore, it has a similar sample size to previously published data (Taylor et al Nature 2021, Firestone et al Scientific Reports 2021). We agree that the total number of tumor are important and have provided that information in **Fig. S8**.

9. Fig 4E, 4F, 4G - the authors should show % of positive cells per mouse instead of them per polyp. The statistical analysis strategy needs to be revised. Please, update these graphs to match the number of mice in each group.

The % positive cells per polyp, rather than per mouse, was the chosen metric to better capture the heterogeneity in polyp size that occurs within each mouse. Representative images are shown per treatment group to convey the spatial distribution of the respective immune cells, which we believe to be the significant result versus absolute quantification as the field aims to induce effective infiltration of anti-tumor immune cells to overcome immune-exclusion (i.e. ineffectual immune cells on the tumor periphery).

10. Fig 4E, 4F - the authors should perform an in-situ flow cytometry analysis to characterize their CD8 population properly. These results are questionable as they are now. Fig 4G needs significance to the proposed mechanism.

Similar to our response to Reviewer 1, while flow cytometry would be more readily quantitative, separation of diseased and healthy tissue is difficult at the macroscopic level in the *APC^{Min}+* model where adenoma presence is variable and present throughout the intestine. Furthermore, tissue dissection and digestion for flow cytometry results in a loss of spatial information that is retained using immunohistochemical methods.

Supplementary figures. Overall, they need updating and grouping following their citing order in the manuscript. There is no need for single images to be shown as figures when the latter is expected to be built as highly complex panels.

We thank the reviewer for this suggestion and have updated both our main and supplementary figures in the Revised manuscript.

Reviewer #3 (Remarks to the Author): with expertise in synthetic biology

Revision nature comms.

In this study, the authors demonstrate that probiotic EcN can be used in mice to monitor for intestinal lesions such as adenoma. Moreover, by using EcN as a frame, the authors can release several therapeutic molecules in the intestine, which greatly reduces the size of tumors.

- 1. Although the results of this work are relevant to the field, several aspects must be taken into consideration:**

The authors claim that "This phenomenon of neoplasia selective colonization of EcN suggested its utility as a platform to both monitor adenoma presence and also then to manipulate lesions in situ." Colonization of lesions and modulation of inflammation with EcN have already been reported in hepatic lesions (doi.org/10.1080/19490976.2021.1924261). In

this study, the authors show that this phenomenon of colonizing lesions and modulating inflammation also happens in the intestine.

• Are there known mechanisms that enable EcN to colonize intestinal lesions and not colonize a healthy intestine? Authors should address this issue in the manuscript.

We are unable to access the paper link provided by the Reviewer, but have included additional text in the revised manuscript to address what is currently known about mechanisms underlying EcN colonization and have added additional citations (Ribet et al *Microbes Infect.* 2015 and Croxen et al *Nature* 2009, Abet et al *Cell Host Microbe* 2016).

Lines 283-289: Equally, selective colonization of neoplastic tissue in humans after oral administration of EcN warrants further investigation in expanded cohorts. In addition, while mechanisms such as selective adhesion to tumor antigens and attraction to small molecules produced in tumors that drive translocation of bacterial pathogens across the gastrointestinal barrier, are known for other E coli strains, Salmonella typhimurium and Fusobacterium nucleatum, a deeper understanding of characteristics governing EcN establishment within tumors is needed.

• The use of EcN expressing luxCDABE allows monitoring the presence of intestinal lesions such as adenomas thanks to the ability of EcN to colonize these lesions. However, the reported data do not show that colonization of intestinal lesions is restricted to adenomas or carcinomas. What are the limits of this method for identifying adenomas and carcinomas and distinguishing them from other types of intestinal lesions that are not linked to colon cancer? Authors should discuss in the text the limitations of this approach for discrimination between different types of lesions.

We thank the Reviewer for the opportunity to clarify this point. In the revised manuscript, we expand on existing data to now include an orthotopic model of microsatellite instability (MSI) CRC in addition to the microsatellite stable (MSS) polyp and CRC models, i.e. broadening the range of CRC subtypes included in the study and showing lesions of both MSS and MSI subtypes are specifically colonized by EcN (Revised Figure 2A-H). We also include reference to existing data describing the colonization potential of EcN in the normal gut and inflammatory lesions. This data suggests that other inflammatory lesions in the gut are not colonized by EcN and hence EcN-based signals are likely to indicate the presence of adenomas or cancer. Additional text and citations have been added to the discussion section of the revised manuscript.

Lines 274-283: In the absence of neoplasia and consistent with our results, orally delivered EcN is not a persistent colonizer of the healthy mouse or human gut, nor inflammatory lesions present in a mouse model of inflammatory bowel disease. Early biodistribution studies with intravenously administered EcN reported that while EcN transiently localized to skin wounds in mice, colonization was not observed at inflamed cutaneous sites or once wounds had healed, suggesting that persistent colonization is a specific feature of neoplasia.

While we explored colonization of EcN across multiple murine models both with intact and disrupted microbiomes, more investigation into the effects of the native tumor microbiome and gut disorders

on EcN colonization could provide insight into generalizability of this approach across patients with varying symptoms, diets and microbiota, including those who might already have baseline levels of EcN present.

2. • For this reviewer, the benefits of detecting EcN by detecting salicylate in urine relative to detecting EcN in fecal samples are not clear. Although it is true that there are easy-to-use commercial kits for detecting salicylate in the urine, there are a number of problems with the use of salicylate to identify the presence of EcN in the intestine after having administered the probiotic. In a possible application in humans maybe salicylate is not the best option. Levels of salicylate in the urine can vary considerably from person to person depending on aspects such as diet or blood pH, among others. This may limit the use of salicylate as an indicator of colonization of intestinal lesions by EcN. Fecal samples appear to be far more reliable by counting the CFU/ml present. The authors should highlight the limits that the use of salicylate as a detection molecule could have for future applications in uncontrolled systems (for example in human applications).

Why do the authors decide to use salicylate?

We thank the Reviewer for the opportunity to clarify why salicylate was chosen as the diagnostic molecule. In Australia the primary screening modality is stool based FOBT. Most simply we wish to move away from stool based testing for new tests we develop, as a requirement to manipulate stool is a key reason given behind non-participation in stool based screening (Jones et al Am. J. Prev. Med. 2010), with blood or even urine based screening preferable for participants. We acknowledge the variability of salicylate at baseline in humans and suspect that it would be important to have patients limit salicylate use (in the form of aspirin or other derivatives) prior to using our technology as a diagnostic. Salicylate has additional therapeutic benefits, suggesting the use of our technology as a theranostic. Additional text regarding therapeutic salicylate has been added to the introduction section of the revised manuscript and to relevant results sections. (Lines: 84-86; 191-195)

3. • Regarding the proposed immunotherapy, based on the release of nanobodies blocking PD-L1 and CTLA-4 targets and cytokines like GM-CSF there are several aspects to discuss:
 - a. Why was it chosen to induce cell lysis to release therapeutic molecules into the extracellular environment instead of using one of the endogenous secretory pathways in EcN?
Cell lysis induced by the expression of a lysis gene (phiX174E) controlled by a circuit as shown in Figure 4a implies a very significant reduction of the colonizing EcN population when AHL levels exceed a certain critical threshold. This is contradictory to one of the aspects that the authors highlight as positive of the use of EcN, i.e. the ability to colonize the lesions and remain in them for long periods of time. The authors must justify the choice of this strategy despite the reduction in EcN colonization.

This is a great question and something we have explored in previous work (Din et al Nature 2016, Gurbatri et al Science Translational Medicine 2020). While constitutive expression is simpler, the lysis circuit serves several major purposes: a) quorum-sensing is only activated in tumors, adding

specificity, b) lysis allows for effective release of therapies and is critical for efficacy, lysis serves as a biocontainment mechanism, keeping the population low and c) bacterial lysis results in immune adjuvants that further enhance therapeutic effects of immunotherapy (Gurbatri et al 2020, Chowdhury et al 2019). Furthermore, we have previously shown that the reduced population due to the lysis circuit results in improved health of mice (Din et al Nature 2016). To address this concern in the context of the manuscript, we have added additional text to highlight this previous work and rationale for choosing this delivery mechanism.

Lines 233-241: We combined therapeutic delivery with an EcN-lux strain genomically-encoded with a lysis circuit optimized (SLIC) to maximize immunotherapeutic release and also aid in biocontainment by controlling EcN populations. Drawing upon previous work, use of this lysis-based release mechanisms is need for effective release of therapies and is critical for therapeutic efficacy. Furthermore, bacterial lysis results in immune adjuvants that further enhance therapeutic effects of immunotherapy. SLIC was used to deliver nanobodies blocking PD-L1 and CTLA-4 targets and cytokines like GM-CSF (SLIC-3), which we have previously demonstrated enhance efficacy of checkpoint blockade therapy in a subcutaneous mouse colorectal model when delivered intratumorally.

b. The circuit in Figure 4a is a circuit based on quorum sensing. The figure is incomplete because this circuit requires expression of the LuxR receptor protein (authors should indicate the production of this protein in Figure 4a). These quorum sensing based circuits exhibit a very ON/OFF response. When the bacterial population exceeds some critical density, the expression of the lysis gene increases dramatically. The critical density required to activate the lysis circuit depends not only on the density of EcN but also, and very decisively, on the expression levels of the LuxR protein (DOI: 10.1093/nar/gku964). Different levels of LuxR expression may imply changes in orders of magnitude in lactone levels (AHL) and hence in the EcN concentration required to activate lysis gene expression.

I wonder how active is the lysis circuit according to the LuxR expression levels in the particular circuit used?

Authors must provide information on the strength of the promoter and RBS under which LuxR is expressed. It is also necessary to know the strength of the RBS upstream the phiX174E gene is expressed.

An in vitro characterization of this circuit (transfer function) is necessary to show the relationship between the density of EcN and the expression levels of phiX174E.

The lysis circuit used in this publication has been used extensively in previous references (Din et al 2016, Gurbatri et al 2020, Chowdhury et al 2019, Savage et al 2023, Vincent et al 2022). In depth characterization of this circuit in the in silico, in vitro, and in vivo context can be found in these citations also provided in the revised manuscript. Briefly, a bacterial population lyses once a critical density or quorum is reached, effectively releasing its therapeutic payload. The quorum-sensing *plux* promoter drives transcription of the quorum-sensing gene, *luxI*, and the phage-derived lysis gene, $\phi X174E$. This circuit has been genomically-integrated, offering stability and also as we've previously shown, enhances therapeutic release compared to plasmid-variants of varying copy numbers. These data were gathered and validated through plate reader assays and in vivo studies comparing copy number variants.

- 4. How the authors can be sure that these levels of concentration of EcN in colonized areas are enough to be sure that the lysis of EcN takes place significantly in the intestine?**

This could have been easily monitored by determining the quantity of CFU/ml present in fecal samples. A significant reduction in the presence of EcN in faecal samples would be a clear indication that EcN lysis is occurring in the gut.

Based on the data provided by the authors, it is not possible to determine the effectiveness of the applied lysis mechanism.

We thank the reviewer for the suggestion. Due to the heterogeneity of the adenomas, it's unlikely that the bacteria would synchronously lyse across all adenomas at the same time. Instead, synchronized lysis will happen when bacteria reach a critical density in individual adenomas. To that end, measuring EcN presence in the fecal samples which are a measure of broad whole intestinal colonization, may not be an accurate representation of lysis. Revised Fig. 1E are histology sections from mice treated with SLIC-PDL1nb tagged with HA, which suggest that the payload is indeed being released intra-adenomally.

- 5. According to the above arguments, surprisingly, the reported data show that improvements in mice with EcN-SLIC are close to those achieved with SLIC-3 (Figure S8). However, assuming the release of therapeutic molecules after EcN lysis, one would expect the impact to be much more significant. This would be compatible with the fact that only a partial lysis of the EcN population may take place. Probably because the densities of EcN are insufficient to fully activate the lysis circuit. These findings suggest that only the probiotic effect of EcN can be sufficient to achieve a notable improvement in terms of tumor area reduction. Of course, the eventual partial release of therapeutic molecules enhances the results. Why do the authors not address in the text the improvements achieved with EcN-SLIC and concentrate only on the results of EcN-SLIC3? I wonder if these improvements would not be even greater if the expression of the lysis gene were done in an inducible system (for instance induced by molecule that can be supplied to mice) and not depending on cell density.**

Based on Figure 4B and C and S8 we do not observe statistically significant differences between the untreated and EcN-SLIC groups. However, we agree that treatment with the EcN-SLIC does trend toward decreased tumor burden. Data shown in Fig. 4E-G and our previous studies (Gurbatri et al 2020, Chowdhury et al 2019), suggest that EcN alone can elicit an immune response resulting in therapeutic benefit. We have added additional text (see below) in the discussion to clarify this point. Regarding an inducible system, such systems lack the benefits offered by our autonomous lysing circuit. As mentioned in previous responses, lysing delivery mechanisms offer population control and biocontainment needed for the clinical translation of these therapeutic platforms.

Lines 250-253: There is a trend whereby SLIC alone has increased granzymeB staining when compared to untreated mice, likely due to the inherent immunogenicity of lysed bacteria^{33, 52}

- 6. Finally, one last minor question is what antibiotic authors use to select EcN on LB plates. Is it resistance that has been introduced in the EcN strain or is it natural EcN resistance to antibiotics like erythromycin or vancomycin?**

EcN-lux was plated on LB plates to assist in selecting for genetically engineered EcN strains and minimize contamination. Er resistance is conferred by the luxCDABE cassette integration.

REVIEWER COMMENTS

Reviewer #1 (Remarks to the Author):

The authors have addressed most, although not all, issues raised. The manuscript has been improved significantly. This is an important work which is worth a rapid publication.

Reviewer #2 (Remarks to the Author):

The authors should clearly state which immunotherapy their engineered EcN delivers into tumors in the abstract.

As previously mentioned and confirmed by the author's rebuttal: The dataset shown in Figure 3 requires a larger N per group of independent experiments. Reproducibility is questionable. The authors should carefully work on it.

As previously mentioned and confirmed by the author's rebuttal: The dataset shown in Fig 4B and 4C needs some improvement. Reproducibility is questionable as group numbers are small and male and female mice are mixed within a given group. Please, keep independent groups for male and female mice.

As previously mentioned and confirmed by the author's rebuttal: Fig 4E, 4F, 4G - the authors should show % of positive cells per mouse instead of them per polyp. As it currently stands, reproducibility needs to be revised since the authors' statistical strategy adds bias to the results and needs to be changed. Please, update these graphs to match the number of mice in each group.

To assist the authors in performing one of my suggestions (Fig 4E, 4F - the authors should perform an in-situ flow cytometry analysis to characterize their CD8 population properly), here is a report dealing with the issue of limited samples (10.1016/j.celrep.2017.03.037). Currently, other methods are available: 10x genomics (<https://www.10xgenomics.com/products/spatial-gene-expression>), MACSima™ Platform (<https://www.miltenyibiotec.com/CA-en/products/macs-imaging-and-spatial->

biology/macsim-

platform.html?query=:relevance:allCategoriesOR:10000717%23OnJlbGV2YW5jZTphbGxDYXRlZ29yaWVzT1I6MTAwMDA3MTc=) or, even, the Columbia University Irving Medical Center (<https://sharedresourcesentrysite.industryrelations.columbia.edu/content/human-immune-monitoring>). Moreover, the authors should determine the co-localization of EcN and immune cells infiltrating the tumors.

Reviewer #3 (Remarks to the Author):

The authors have addressed all questions. In my opinion, this article is well-suited for publication.

Reviewer #4 (Remarks to the Author):

In this paper, Gurbatri and co-authors, investigate how *E. coli* Nissle 1917 (EcN) can selectively colonise premalignant and malignant colorectal lesions and how it can be used as a diagnostic and therapeutic tool in CRC. To do so, they developed different mouse models and genetically modified EcN strains, as well as a small controlled clinical trial :

- Firstly, they show that there is selective tropism of EcN in tumour and pre-tumour tissues in different mouse models of CRC, independent of colibactin expression (in APC min/+ mice and orthotopic models of MSI and MSS CRC). Despite some limitations (see below), the authors show similar results in humans based on the results of a small clinical trial evaluating tumour colonisation in CRC in surgical patients after 14 days of oral administration of a previously marketed, unmodified EcN strain (Mutaflor).

- Secondly, Gurbatri et al demonstrate that the modified EcN strain can be used to distinguish APCmin/+ mice from controls using both microbiological detection tests from faeces and salicylate production detected in urine.

- Thirdly, genetically modified EcN strains that locally deliver immune checkpoint inhibitors and GM-CSF with an optimised lysis circuit can trigger an antitumour response associated with increased T cell infiltration in APCmin/+ mice.

From a scientific and clinical point of view, this work explores in an original way unmet medical needs with elegant and promising approaches that open the way to bacterial engineering for the detection and treatment of colorectal cancer. Significant improvements have been made in response to comments from other reviewers. The structure of this revised version is easy to follow and read and the main results are correctly introduced and highlighted.

Key comments relating in particular to the clinical trial :

In general, although it provides significant insight into the potential application to humans of the approach explored in this article, the small size of the clinical trial added and the heterogeneity of results among patients treated with Mutaflor raise questions about the possibility of drawing conclusions from the data as presented:

- To quantify EcN colonisation after oral administration of EcN (Mutaflor) in CRC patients, the authors designed a specific approach, with new nested PCR strategy with multiple steps and culture amplification to overcome multiple technical problems that notably resulted in the loss of a significant number of samples (n=15 patients, Line 396). For internal validation, it would have been interesting to confirm the qPCR results by immunofluorescent staining as the authors did in mice (if possible, samples from the previous 15 patients who were excluded from the analysis may also be added to increase statistical power). Especially as most of the normal colon samples from patients treated with either placebo or Mutaflor were below the limit of detection (figure 2K), the use of another method of quantification would be of great importance.

- Two out of 6 patients (33%!) treated with Mutaflor did not show any specific enrichment between normal colon and tumour (Figure 2K - upper dots), which should be discussed. This also raises questions about whether conclusions can be drawn from such a limited number of patients. In particular, some confounding factors should be described:

- o Are there differences in treatment uptake between participants that could be responsible for EcN enrichment in the tumour? How was treatment uptake controlled and monitored?

o As the authors have shown in mice, the interval between the last probiotic intake and sampling can modify the transient concentration of EcN in faeces and very likely in tissues. What was the interval between the last dose and the surgical procedure? This should be added to the methods. Was this time consistent between all patients?

o In the inclusion criteria of the attached trial design, patients were eligible even if they had been treated with an antibiotic in the weeks prior to the start of the trial. As antibiotics can modulate the ability of probiotics to colonise the gut, it would be important to add this parameter to Table S1 describing the patients (from the protocol of the clinical trial, this parameter has been collected).

- The authors mention in the methods section that the clinical trial was stopped prematurely due to concerns about colibactin-expressing *E. coli* being procarcinogenic. Colibactin has been known to have potential procarcinogenic effects for many years prior to the start of this trial (PMID: 29420293, PMID: 22903521, PMID: 23846483) and the publication cited by the authors (Pleguezuelos-Manzano et al, Nature 2020). These available data did not prevent ethical approval (ref. HREC/18/CALHN/751). Why did this safety issue arise during the trial? Was there a clinical signal during the trial that raised this question?

This early termination should at least be mentioned in the results section to explain to the reader why there were so few points available in Figure 2K. This concern should also be discussed later highlighting the need to control and prevent the potential pro-carcinogenic effect of unmodified EcN or other probiotic strains when considering future therapeutic and diagnostic applications.

- Minor comments:

o For Patient RAP13, the CRC stage (Table S1) did not mention the type of classification used. This should be clarified (pTNM or cTNM).

We would like to thank the reviewers for further feedback that allowed us to clarify the data and manuscript conclusions. Briefly, in this revised version, we have included our methodology for our automated IHC staining analysis and additional text in our manuscript to address study limitations.

REVIEWER COMMENTS

Reviewer #1 (Remarks to the Author):

The authors have addressed most, although not all, issues raised. The manuscript has been improved significantly. This is an important work which is worth a rapid publication.

Reviewer #2 (Remarks to the Author):

The authors should clearly state which immunotherapy their engineered EcN delivers into tumors in the abstract.

We have made this change and added the following revised sentence to the abstract (below in red):

Line 50 - To assess therapeutic potential, we engineer EcN to locally **release a cytokine, Granulocyte-Macrophage Colony Stimulating Factor (GM-CSF) and blocking nanobodies against programmed cell death protein – ligand 1 (PD-L1) and cytotoxic T- lymphocyte-associated pro-teiin-4 (CTLA-4)** at the neoplastic site, and demonstrate that oral delivery of this strain reduces adenoma burden by ~50%.

As previously mentioned and confirmed by the author's rebuttal: The dataset shown in Figure 3 requires a larger N per group of independent experiments. Reproducibility is questionable. The authors should carefully work on it.

To address the concern for reproducibility, we have included all previously run trials across distinct cohorts as supplementary figure, **Fig. S8.** and have included text in the discussion (and below in red) addressing smaller cohort sizes. Taken together, these cohorts demonstrate increased salicylate or its primary metabolite salicylic acid levels in tumor-bearing mice compared to WT control mice.

Line 280 - **Finally, evaluation in expanded cohorts for both screening and diagnostic applications is needed prior to translation of these technologies.**

As previously mentioned and confirmed by the author's rebuttal: The dataset shown in Fig 4B and 4C needs some improvement. Reproducibility is questionable as group numbers are small male and female mice are mixed within a given group. Please, keep independent groups for male and female mice.

We have distinguished male and female mice in the figure by colour and have included these details in the legend. We have not observed any evidence of sex-specific phenomenon, which supports our primary aim of generating data relevant to a population containing both sexes prior to clinical studies. Furthermore, we ensured that the phenomenon was not due to cohort-specific effects by aggregating our data across independent cohorts.

As previously mentioned and confirmed by the author's rebuttal: Fig 4E, 4F, 4G - the authors should show % of positive cells per mouse instead of them per polyp. As it currently stands, reproducibility needs to be revised since the authors' statistical strategy adds bias to the results and needs to be changed. Please, update these graphs to match the number of mice in each group.

We thank the reviewer for the opportunity to clarify our analysis. We used an automated IHC quantification method with minimal user input to minimize bias. Briefly, this algorithm thresholds all images and counts the puncta per polyp and reports the % area per polyp. We have included intermediate images as a supplementary figure, Fig. S10A, to demonstrate this process and improve transparency of this analysis. We have additionally included analysis of % positive cells per mouse as a supplementary figure, Fig. S10B and below.

Fig. S10 (A) Automated analysis of IHC staining where images represent (top left) manual segmentation of adenoma, (top right) deconvoluted image, (bottom left) thresholded image, (bottom right) measuring of the foreground pixels. (B) Related to Fig. 4E-G, the total % positive count per mouse.

To assist the authors in performing one of my suggestions (Fig 4E, 4F - the authors should perform an in-situ flow cytometry analysis to characterize their CD8 population properly), here is a report dealing with the issue of limited samples (10.1016/j.celrep.2017.03.037). Currently, other methods are available: 10x genomics (<https://www.10xgenomics.com/products/spatial-gene-expression>), MACSima™ Platform (<https://www.miltenyibiotec.com/CA-en/products/macs-imaging-and-spatial-biology/macsima-platform.html?query=:relevance:allCategoriesOR:10000717%23OnJibGV2YW5jZTphb>)

GxDYXRIZ29yaWVzT1I6MTAwMDA3MTc=) or, even, the Columbia University Irving Medical Center (<https://sharedresourcesentrysite.industryrelations.columbia.edu/content/human-immune-monitoring>).

For flow cytometry, to ensure we are collecting neoplastic tissue for analysis, we only dissect overt, macroscopic disease prior to fresh tissue dissociation for cell staining. Previously we have undertaken flow cytometry analysis after SLIC-treatment using a subcutaneous tumor model that generates much larger neoplastic tumor tissue volume to work with (Gurbatri et al., Sci Translational Med 2020) and observed an increase in activated CD8+ T cells (CD8+IFN γ +TNF α +). For the current study we deliberately used orthotopic models to maximise translational relevance of this work. It is difficult however to generate reliable cytometry data on these small amounts of tissue from the orthotopic *Apc^{min}* model, without using (unconscionably) large cohorts of mice. Our approach is to minimise animal usage by analysis of sections from tissue samples which also has the side benefits of enabling us to locate both macroscopic and microscopic disease and preserves spatial information about the location of lymphocytes in the sample.

We thank the Reviewer for understanding these limitations of flow cytometry analysis on small tissue samples and providing these resource suggestions. These recent developments in spatial transcriptomics and multiplexed immunohistochemical platforms are exciting and enable characterisation of discrete cell populations within tissue samples. However, they have primarily been developed for human tissue samples (using antibodies to recognise human epitopes or the human transcriptome), or at lower than single-cell resolution (eg. current standard release of 10x visium platform). Other similar technologies such as the MERSCOPE (single molecule FISH) require de novo characterisation of bespoke gene panels. Of particular relevance to our study, in which we have formalin-fixed, paraffin embedded (FFPE) mouse tumour tissue samples for analysis, standard anti-mouse multiplex immuno-stain protocols to analyse the tumor microenvironment commonly require non-standard, paraformaldehyde fixed/cryosectioned tissue (Miltenyi Biotec MICS) and/or are not very sophisticated and are unlikely to provide more CD8 information than we have already obtained from our single-plex immunostaining (eg Maxpar Mouse Immuno-Oncology IMC Panel, Fluidigm). We appreciate the technological and spatial platform advances that will soon enable more refined characterisation of the CD8 T cell population, particularly in size limited mouse tumor samples such as ours. Currently though, it would be a significant undertaking in itself to setup one of these platforms and we feel it is outside the scope of the present study.

Moreover, the authors should determine the co-localization of EcN and immune cells infiltrating the tumors.

Unfortunately, we do not have these samples available to co-stain, but multiple studies have explored EcN and immune cell infiltration (Zhang et al Applied and Environmental Microbiology 2012, Luke et al Clinical Cancer Research 2023). The first study listed, demonstrates increased neutrophil presence in B16 and 4T1 tumors upon EcN treatment. The second explores the use of engineered probiotic EcN in clinical trial trials of refractory advanced cancers and noted that patients treated with their EcN resulted in increased infiltration of CD8 and CD4 T cells intratumorally. Given that we also see a similar pattern of immune cell infiltration, and immune activation post bacterial colonization is well-known, it is likely that EcN is the driving factor underlying this phenomenon.

Reviewer #3 (Remarks to the Author):

The authors have addressed all questions. In my opinion, this article is well-suited for publication.

Reviewer #4 (Remarks to the Author):

In this paper, Gurbatri and co-authors, investigate how *E. coli* Nissle 1917 (EcN) can selectively colonise premalignant and malignant colorectal lesions and how it can be used as a diagnostic and therapeutic tool in CRC. To do so, they developed different mouse models and genetically modified EcN strains, as well as a small controlled clinical trial :

- Firstly, they show that there is selective tropism of EcN in tumour and pre-tumour tissues in different mouse models of CRC, independent of colibactin expression (in APC min/+ mice and orthotopic models of MSI and MSS CRC). Despite some limitations (see below), the authors show similar results in humans based on the results of a small clinical trial evaluating tumour colonisation in CRC in surgical patients after 14 days of oral administration of a previously marketed, unmodified EcN strain (Mutaflor).
- Secondly, Gurbatri et al demonstrate that the modified EcN strain can be used to distinguish APCmin/+ mice from controls using both microbiological detection tests from faeces and salicylate production detected in urine.
- Thirdly, genetically modified EcN strains that locally deliver immune checkpoint inhibitors and GM-CSF with an optimised lysis circuit can trigger an antitumour response associated with increased T cell infiltration in APCmin/+ mice.

From a scientific and clinical point of view, this work explores in an original way unmet medical needs with elegant and promising approaches that open the way to bacterial engineering for the detection and treatment of colorectal cancer. Significant improvements have been made in response to comments from other reviewers. The structure of this revised version is easy to follow and read and the main results are correctly introduced and highlighted.

Key comments relating in particular to the clinical trial :

In general, although it provides significant insight into the potential application to humans of the approach explored in this article, the small size of the clinical trial added and the heterogeneity of results among patients treated with Mutaflor raise questions about the possibility of drawing conclusions from the data as presented:

- To quantify EcN colonisation after oral administration of EcN (Mutaflor) in CRC patients, the authors designed a specific approach, with new nested PCR strategy with multiple steps and culture amplification to overcome multiple technical problems that notably resulted in the loss of a significant number of samples (n=15 patients, Line 396). For internal validation, it would have been interesting to confirm the qPCR results by immunofluorescent staining as the authors did in mice (if possible, samples from the previous 15 patients who were excluded from the analysis may also be added to increase statistical power). Especially as most of the normal colon samples from patients treated with either placebo or Mutaflor were below the limit of detection (figure 2K), the use of another method of quantification would be of great importance.

Thank you for your comments.

Our method to specifically localise EcN-lux in mouse samples using RNAscope relies on detection of the lux-cassette, i.e. is specific for the genetic modification we have introduced into EcN-lux in a coding gene to generate *lux* RNA, which is what the RNAScope probe is detecting (Fig 2E).

We are not able to transfer this approach to the human clinical trial samples as participants in the trial were administered non-genetically modified EcN (Mutaflor), i.e, the probiotic bacteria do not express *lux*. We have analysed the genome of *E coli* Nissle, in comparison to closely related gut commensal *E.coli* strains, and have been unable to find transcribed regions of the genome that can differentiate these related strains (to then target using a similar RNAscope approach). We were able to isolated genomic regions that allow us to detect specific EcN DNA sequences by PCR (Fig 2K), but these regions are not transcribed, so not amenable to RNAscope analysis using FFPE patient tissues.

We have undertaken pan-gram negative bacteria immunofluorescence staining using anti-LPS antibodies (as in Fig 2F on mouse tissue) to localise gram-negative bacteria in tissue samples from participants in the clinical trial. From this analysis we can clearly observe pockets of bacterial colonisation on the luminal surface of the tissue samples, but no clear differences in colonisation between samples from patients in the placebo or Mutaflor groups, or intra-patient normal colorectum to tumour comparisons. This is understandable, as we would expect to detect bacteria in normal colon and tumor samples, in both placebo and Mutaflor trial participants. We have included a representative image of this staining in Supp Fig 4 and also included additional images here for the reviewer (Figure for reviewer below). In the absence of an alternate specific, and highly sensitive, staining method for non-genetically modified *E coli* Nissle, we are not able to include a second method of quantification. We now make this limitation clear in the discussion (new text in red).

Line 299 - *The limitations of our current trial...in the absence of other highly sensitive and EcN-specific markers to enable tissue localisation the detection of non-genetically modified EcN was only able to be assessed using PCR;*

Figure for reviewers. Visualizing gram negative bacteria in human CRC. (A) Representative images from histopathological analysis of tumor tissue (A-D & I-L) and normal colon (E-H & M-P) resected from CRC patients (placebo n = 2 and Mutaflor n = 5 patients). Serial sections of tumor (A-D, also shown in Supp Fig 4) and normal colon (E-H) from patient treated with placebo, stained with (A, E) H&E, immunofluorescent stain no primary control (B, F) or against lipopolysaccharide (LPS, green, C, D, G, H). Serial sections of tumor (I-L) and normal colon (M-P) from patient treated with Mutaflor, stained with (I, M) H&E, immunofluorescent stain with no primary control (J, N) or against lipo-polysaccharide (LPS, green, K, L, O, P). Scale bars A, E, I & M 100 μ m, D, H, L & P 10 μ m.

- Two out of 6 patients (33%!) treated with Mutaflor did not show any specific enrichment between normal colon and tumour (Figure 2K - upper dots), which should be discussed.

We have added text to the discussion as follows (new text in red).

Line 291 - *Equally, in our trial two of the six participants in the Mutaflor arm of the trial had comparatively high levels of EcN in the normal, adjacent colon samples and did not show EcN enrichment in tumour samples over normal. As the stool CFU data from our mouse models (Fig. 3B, S6) indicate that individual mice show variability in the time required to clear EcN from their gastrointestinal tract in the absence of neoplasia after EcN dosing, this result in our human cohort may be due to the absence of a multi-day wash-out period in our trial protocol and/or inter-patient variability in colonisation by probiotic bacteria (Zmora et al., 2018 Cell)*

This also raises questions about whether conclusions can be drawn from such a limited number of patients. In particular, some confounding factors should be described:

o Are there differences in treatment uptake between participants that could be responsible for EcN enrichment in the tumour? How was treatment uptake controlled and monitored?

o As the authors have shown in mice, the interval between the last probiotic intake and sampling can modify the transient concentration of EcN in faeces and very likely in tissues. What was the interval between the last dose and the surgical procedure? This should be added to the methods. Was this time consistent between all patients?

We agree and have added the following text to the methods to clarify these important points-

Methods line 411 - Participants were provided with probiotic tablets and instructed to begin 14 days prior to resection. On the morning of resection investigators verbally confirmed with the participants that they had taken the entire probiotic course, stopping on the day before surgery. There were no alternate methods employed to validate treatment uptake, other than PCR detection of EcN DNA sequence in microbial cultures from tissue samples.

o In the inclusion criteria of the attached trial design, patients were eligible even if they had been treated with an antibiotic in the weeks prior to the start of the trial. As antibiotics can modulate the ability of probiotics to colonise the gut, it would be important to add this parameter to Table S1 describing the patients (from the protocol of the clinical trial, this parameter has been collected).

Only patients that did not take probiotics or antibiotics during the study period were recruited. However, information on prior antibiotic usage before the study period was not reliably, prospectively collected. Unfortunately we cannot retrospectively validate prior antibiotic usage in participants as we can only access patient medical records within our hospital health network, not within external clinical practises where they are can also be prescribed. As such, we have amended our study protocol for publication to remove the sentence 'As antibiotic pre-treatment does affect probiotic colonisation, antibiotic usage will be recorded in our database.' We now also make this lack of prior antibiotic usage information clear in our limitation section in the discussion (new text in red).

Line 302 - *The limitations of our current trial include: small participant numbers; incomplete previous antibiotic usage data; a reliance on self-reporting of probiotic administration by trial participants; in the absence of other highly sensitive and EcN-specific markers to enable tissue localization, the detection of non-genetically modified EcN was only able to be assessed using PCR; lastly, a wash-out period post-probiotic treatment, but prior to tissue resection, was not included but will be key to understanding the longevity of EcN neoplastic colonisation.*

- The authors mention in the methods section that the clinical trial was stopped prematurely due to concerns about colibactin-expressing E. coli being

procarcinogenic. Colibactin has been known to have potential procarcinogenic effects for many years prior to the start of this trial (PMID: 29420293, PMID: 22903521, PMID: 23846483) and the publication cited by the authors (Pleguezuelos-Manzano et al, Nature 2020). These available data did not prevent ethical approval (ref. HREC/18/CALHN/751). Why did this safety issue arise during the trial ? Was there a clinical signal during the trial that raised this question?

No, there was no clinical signal that arose during the trial. Our closure of the trial was in response to new data that was published in early 2020 when patient recruitment was occurring.

We agree that pks+ *E. coli* were proposed to have potential DNA-damaging, pro-carcinogenic effects prior to the start of this trial, however, Mutaflor (probiotic with active agent *E. coli* Nissle, a strain of pks+ *E. coli*) was approved for clinical use worldwide and had been used in humans for over a century. What was unclear was the possible increased risk upon administration of pks+ *E. coli* (such as EcN) to participants given: the diverse microbiome already found in humans (the epidemiological data for pks+ *E. coli* with CRC was associative and in fairly small cohorts only); our and other mouse data and earlier clinical studies suggested that EcN is not a good long-term coloniser of the normal gut (PMID: 35931082, PMID: 19637333); how the EcN strain fit within the broader range of invasive, adherent pks+ *E. coli* strains that had been studied in the papers cited by the reviewer and PMID: 24658599; or compelling data that EcN is a key etiologic agent for CRC or makes the genotoxin colibactin *in vivo*.

In Feb 2020, compelling functional data on the etiologic role of colibactin in colorectal cancer was published (PMID: 32106218, supported by PMID: 32483361 in July 2020). This included identification of the precise DNA mutation signature generated in gut epithelial cells by exposure to pks+ bacteria, that could then be tracked back to 6-8% of all CRCs, including in key tumour suppressor genes for CRC, in a large dataset of over 5000 cases.

This was of new relevance, particularly as we were recruiting patients in the colonoscopy screening group – where most patients do not have colorectal cancer. We recognise that while Mutaflor has been used in patients for over a century with a solid safety record, that long-term (10-20y) follow-up of patients to investigate possible increased cancer rates in patients on Mutaflor was lacking. Ardeypharm (company that produces Mutaflor) conducted rigorous, but relatively short-term, mutagenesis assays following EcN dosing in rats and was unable to find evidence of genotoxicity *in vivo* after a month, but they did not look for colibactin associated mutational signatures after extended time periods (Dubbert et al 2020 Eur J of Microbiol & Immunol DOI: 10.1556/1886.2019.00025). As such, and in the setting of COVID related reductions in trial activity, we closed this trial early.

This early termination should at least be mentioned in the results section to explain to the reader why there were so few points available in Figure 2K. This concern should also be discussed later highlighting the need to control and prevent the potential pro-carcinogenic effect of unmodified EcN or other probiotic strains when considering future therapeutic and diagnostic applications.

We have amended the text as follows (new text in red):

Results Line 174 - *A small number of participants were recruited to this study after early closure due to COVID-19 related trial restrictions and concerns over colibactin producing E coli³⁴ such as EcN. Homogenates from matched normal and tumor tissue (n = 8 patients) were cultured to enrich for microbial content, DNA was then isolated and subjected to qPCR assays (Fig. S5). Despite the smaller than intended number of samples, EcN-specific PCR*

amplicons^{27, 47} indicated significant enrichment of this bacteria in cultures from tumor tissue in patients administered Mutaflor, but not placebo controls (Fig. 2J-K, Fig. S5C).

Discussion Line 291 - *Equally, in our trial two of the six participants in the Mutaflor arm of the trial had comparatively high levels of EcN in the normal, adjacent colon samples and did not show EcN enrichment in tumour samples over normal. As the stool CFU data from our mouse models (Fig. 3B, S6) indicate that individual mice show variability in the time required to clear EcN from their gastrointestinal tract in the absence of neoplasia after EcN dosing, this result in our human cohort may be due to the absence of a multi-day wash-out period in our trial protocol and/or inter-patient variability in colonisation by probiotic bacteria (Zmora et al., 2018 Cell) As such, selective colonization of neoplastic tissue in humans after oral administration of EcN warrants further investigation in expanded cohorts. Of note, safety concerns regarding EcN, as a possible colibactin-producing E. coli strain, remain. Future trial design should incorporate colibactin-mutant strains that retain neoplastic colonisation properties (Fig. 1). The limitations of our current trial include: small participant numbers; incomplete previous antibiotic usage data; a reliance on self-reporting of probiotic administration by trial participants; in the absence of other highly sensitive and EcN-specific markers to enable tissue localisation the detection of non-genetically modified EcN was only able to be assessed using PCR; a wash-out period post-probiotic treatment, but prior to tissue resection, was not included but will be key to understanding the longevity of EcN neoplastic colonisation. Nevertheless, this new data demonstrates EcN can colonise human colorectal neoplasia and thus supports the findings from our mouse studies.*

- Minor comments:

o For Patient RAP13, the CRC stage (Table S1) did not mention the type of classification used. This should be clarified (pTNM or cTNM).

Apologies for this oversight, all tumour grading was pTNM. We have added this information to the table.

REVIEWERS' COMMENTS

Reviewer #4 (Remarks to the Author):

We thank the authors for addressing all the points we have suggested, clearly presenting the current limitations of the clinical trial, which still bring interesting data and concepts to the field.

All in all, this is a very interesting work that, in my opinion, deserves to be widely disseminated.